# The PAX-FOXO1s trigger fast trans-differentiation of chick embryonic neural cells into alveolar rhabdomyosarcoma with tissue invasive properties limited by S phase entry inhibition

Gloria Gonzalez Curto[1☯], Audrey Der Vartanian[2☯], Youcef El-Mokhtar Frarma[1☯], Line Manceau[1☯], Lorenzo Baldi[1☯], Selene Prisco[1], Nabila Elarouci[3], Frédéric Causeret[4,5], Daniil Korenkov[1], Muriel Rigolet[2], Frédéric Aurade[2,6], Aurélien De Reynies[3], Vincent Contremoulins[7], Frédéric Relaix[2], Orestis Faklaris[7¤], James Briscoe[8], Pascale Gilardi-Hebenstreit[1], Vanessa Ribes[1]*

**1** Université de Paris, CNRS, Institut Jacques Monod, Paris, France, **2** Univ Paris Est Créteil, INSERM, EnVA, EFS, IMRB, Créteil, France, **3** Programme Cartes d'Identité des Tumeurs, Ligue Nationale Contre le Cancer, Paris, France, **4** Université de Paris, *Imagine* Institute, Team Genetics and Development of the Cerebral Cortex, Paris, France, **5** Université de Paris, Institute of Psychiatry and Neuroscience of Paris, INSERM U1266, Paris, France, **6** Sorbonne Université, INSERM, UMRS974, Center for Research in Myology, Paris, France, **7** ImagoSeine core facility of Institut Jacques Monod and member of France-BioImaging, France, **8** The Francis Crick Institute, 1 Midland Road, London, United Kingdom

☯ These authors contributed equally to this work.
¤ Current address: MRI-CRBM imaging facility, CNRS, 34293 Montpellier cedex, France
* vanessa.ribes@ijm.fr

## Abstract

The chromosome translocations generating PAX3-FOXO1 and PAX7-FOXO1 chimeric proteins are the primary hallmarks of the paediatric fusion-positive alveolar subtype of Rhabdomyosarcoma (FP-RMS). Despite the ability of these transcription factors to remodel chromatin landscapes and promote the expression of tumour driver genes, they only inefficiently promote malignant transformation *in vivo*. The reason for this is unclear. To address this, we developed an *in ovo* model to follow the response of spinal cord progenitors to PAX-FOXO1s. Our data demonstrate that PAX-FOXO1s, but not wild-type PAX3 or PAX7, trigger the trans-differentiation of neural cells into FP-RMS-like cells with myogenic characteristics. In parallel, PAX-FOXO1s remodel the neural pseudo-stratified epithelium into a cohesive mesenchyme capable of tissue invasion. Surprisingly, expression of PAX-FOXO1s, similar to wild-type PAX3/7, reduce the levels of CDK-CYCLIN activity and increase the fraction of cells in G1. Introduction of CYCLIN D1 or MYCN overcomes this PAX-FOXO1-mediated cell cycle inhibition and promotes tumour growth. Together, our findings reveal a mechanism that can explain the apparent limited oncogenicity of PAX-FOXO1 fusion transcription factors. They are also consistent with certain clinical reports indicative of a neural origin of FP-RMS.

**Data Availability Statement:** Transcriptomes of ARMS and ERMS biopsies have been published elsewhere (publications cited in our manuscript; accession numbers GSE92689, E-TABM-1202, E-MEXP-121). These are microrrays. The normalized values necessary to be able to compare data coming from distinct labs are also provided in S1 Table.

**Funding:** VR, FC, FR are staff scientists from the INSERM, PGH is a research director of the CNRS, FR is professor from UPEC. LM has obtained a fellowship from University of Paris. Work in the lab of VR was supported by the Ligue Nationale Contre le Cancer grant (PREAC2016.LCC). Work in FR lab was supported by Agence Nationale pour la Recherche (ANR) grant Crestnetmetabo (ANR-15-CE13-0012-02) and Fondation pour la Recherche Médicale (FRM; Grant FDT20130928236). JB is supported by the Francis Crick Institute, which receives its core funding from Cancer Research UK, the UK Medical Research Council and Wellcome Trust (all under FC001051) and the European Research Council (AdG 742138). The funders had no role in study design, data collection and analysis, decision to publish, or preparation of the manuscript.

**Competing interests:** The authors have declared that no competing interests exist.

## Author summary

The fusion-positive subtype of rhabdomyosarcoma (FP-RMS) is a rare malignant paediatric cancer, whose induction and evolution still remain to be deciphered. Out of the gross genetic aberrations found in these cancers, t(2:13) and t(1,13) chromosome translocations are the first to appear and lead to the expression of fusion proteins made of the DNA binding domains of either PAX3 or PAX7 and the transactivation domain of FOXO1. Both PAX3-FOXO1 and PAX7-FOXO1 have a strong impact on gene transcription, yet they only inefficiently promote the transformation of healthy cells into tumorigenic cells. To address this issue, we have used chick embryos to monitor *in vivo* the early response of cells to PAX-FOXO1 chimeric proteins. We showed that both proteins, but not the normal PAX3 and PAX7, transform neural cells into cells with FP-RMS molecular features. The PAX-FOXO1s also force polarized epithelial neural cells to adopt a mesenchymal phenotype with tissue invasive properties. However, the PAX-FOXO1s inhibit cell division and hence tumour growth. Genetically re-activating core cell cycle regulators rescues PAX-FOXO1 mediated cell cycle inhibition. Together, our findings bring further support to the idea that the PAX-FOXO1s are *stricto sensu* oncoproteins, whose oncogenicity is limited by negative effects on cell cycle.

## Introduction

Transcriptomic landscape remodelling represents a hallmark of tumourigenesis [1]. This is often achieved by perturbing the activity of powerful transcriptional modulators, such as master transcription factors (TFs). Understanding how the activity of these factors lead to a pathogenic transformation of cells represents a key challenge in cancer research, so is the development of *in vivo* and more physiological model systems to address this question [1,2].

Two related oncogenic TFs, PAX3-FOXO1 and PAX7-FOXO1, are associated with the emergence and development of the paediatric alveolar subtype of rhabdomyosarcoma (RMS), named fusion-positive RMS (FP-RMS) [3]. Primary lesions in FP-RMS patients are mostly found in limb extremities and the trunk. These comprise aggregates of round cells usually delineated by fibrous septa that express, as for other RMS subtypes, undifferentiated embryonic muscle cells markers. Along with these primary lesions, almost half of FP-RMS patients carry detectable metastases in the lung or bone marrow at the time of diagnosis. The occurrence of these metastases, together with cancer resistance and emergence of secondary disease are to blame for a poor cure rate of FP-RMS patients [4].

The in-frame pathognomonic chromosomal translocations, t(2;13)(q35;q14) or t(1;13) (p36;q14) fuse the 5' end of the *PAX3* or *PAX7* genes to the 3′ end of the *FOXO1* gene and lead to the mis-expression of chimeric TFs made of the DNA binding domains of PAX3 or PAX7 TFs and the transactivation domain of FOXO1 [3]. Exome sequencing revealed that these translocations are the primary genetic lesions in more than 90% of FP-RMS cases [5,6]. Few somatic mutations are found in FP-RMS suggesting a relative fast development of the tumour after the translocations [6]. Furthermore, recurrent gross genetic aberrations, including whole genome duplication, unbalanced chromosomal copy gain, focal amplifications (12q13-q14 amplicon), or loss of heterozygosity notably on 11p15.5 locus presented by FP-RMS cells [5,6] suggest a tumorigenic transformation associated with chromothripsis [7]. The relative contribution of PAX-FOXO1s and of these gross genetic aberrations during the transformation of healthy cells into FP-RMS cells is still debated.

A large body of work, mainly focused on PAX3-FOXO1 and aimed at identifying and functionally characterizing PAX-FOXO1's target genes, argues the cell fate change characteristic of FP-RMS is driven by PAX-FOXO1s [8,9]. This is hypothesised to arise from PAX-FOXO1's strong transcriptional transactivation potential, which surpasses that of normal PAX3 and PAX7 [10–13]. PAX3-FOXO1 binds to non-coding *cis*-regulatory genomic modules (CRMs), remodelling chromatin and enhancing transcriptional activity [11,12]. These CRMs regulate the expression of genes associated with at least 3 traits deleterious for patients' health [8,9,11,12,14]. First, several of the target genes encode cell surface proteins which are key cell migration regulators and the alteration of the some of them was shown to affect RMS cell motility [15–19]. Second, FP-RMS cells express undifferentiated muscle cell master TFs, which in presence of PAX3-FOXO1 can no longer promote muscle terminal differentiation [20,21]. Third, PAX-FOXO1s perturb the core cell cycle machinery [8,9]. Cross-interactions between PAX3-FOXO1 with the anti-apoptotic gene *BCL-XL* or the senescent factor p16$^{INK4A}$ promote cell survival [22–24]. PAX3-FOXO1 increases the proliferation of fibroblasts and myoblasts and this associated with a downregulation of cyclin-dependent kinase inhibitors (CDKN1B and CDKN1C) [25,26]. The fusion protein displays elevated levels in the G2 phase which are required for the upregulation of G2/M checkpoint adaptation genes [27].

Despite the apparently powerful activity of PAX-FOXO1s, data from animal models have led to the conclusion that the fusion proteins do not efficiently trigger FP-RMS formation and spreading [24,28–31]. In excess of 60 days are required for grafted PAX3-FOXO1 expressing human myoblasts or mesenchymal stem cells to produce significant FP-RMS like growths in mice. This contrasts with the 15 days required for patient derived FP-RMS cells [30–32]. Similarly, driving PAX3-FOXO1 expression in the muscle embryonic cells from the murine *Pax3* locus induces tumour mass with a reported frequency of 1 in 228 [13]. Importantly, these *in vivo* approaches have revealed several parameters enhancing PAX-FOXO1 proteins oncogenicity. The random insertion of transgenes in zebrafish indicated that neural derived tissues are more prone than mesodermal derived tissues to produce tumours when exposed to PAX3-FOXO1, highlighting the differential response of distinct cell lineages [29]. In addition, zebrafish and mouse models both indicate that a threshold level of PAX3-FOXO1 needs to be reached to observe tumourigenesis [28,29,32]. Finally, complementing PAX-FOXO1s expression with genetic aberrations promoting cell cycle progression markedly accelerated and increased the frequency of tumour formation [28–34]. This was notably achieved by lowering the expression of p53 or the retinoblastoma protein, RB1; or conversely by ectopically elevating MYCN expression or RAS activity.

To investigate the molecular mechanisms of oncogenicity in FP-RMS we characterised the initial cellular and molecular steps associated with the transformation of cells expressing PAX3-FOXO1 and PAX7-FOXO1. The growing evidence for an embryonic origin of paediatric cancers [35], the identification of FP-RMS growths in neural tube derived tissues [36,37], the recurrent presence of embryonic neural lineage determinants in FP-RMS cells [9], and the recent use of chick embryos to study cancer cells migration and invasion [38,39] led us to develop the embryonic chick neural tube as a model system. We demonstrate that PAX-FOXO1s repress the molecular hallmarks of neural tube progenitors within 48 hours and impose a molecular signature reminiscent of that of FP-RMS cells. Concomitantly, PAX-FOXO1s promote an epithelial-mesenchymal transition, conferring on cells the ability to invade the adjacent mesoderm in less than 72 hours. Moreover, PAX-FOXO1s limit cell cycle progression, *via* a reduction of CDK-CYCLIN activity, which in turn can explain the limited oncogenicity of these fusion TFs.

## Results

### Chick neural cells lose their neurogenic potential upon PAX3-FOXO1 exposure

To investigate the transformation potential of PAX-FOXO1 proteins, we set out to perform gain of function experiments *in vivo* using the neural tube of chick embryos. Hamburger and Hamilton (HH) stage 11 chick embryos were first electroporated with a vector expressing *PAX3-FOXO1* together with a bi-cistronically encoded nuclear-targeted GFP and allowed to develop *in ovo* for up to 72h (Fig 1A). For comparison, electroporations with the wild-type versions of *Pax3* or the empty *pCIG* vector were performed. In addition, the non-electroporated side of the neural tube stood as well as an internal control.

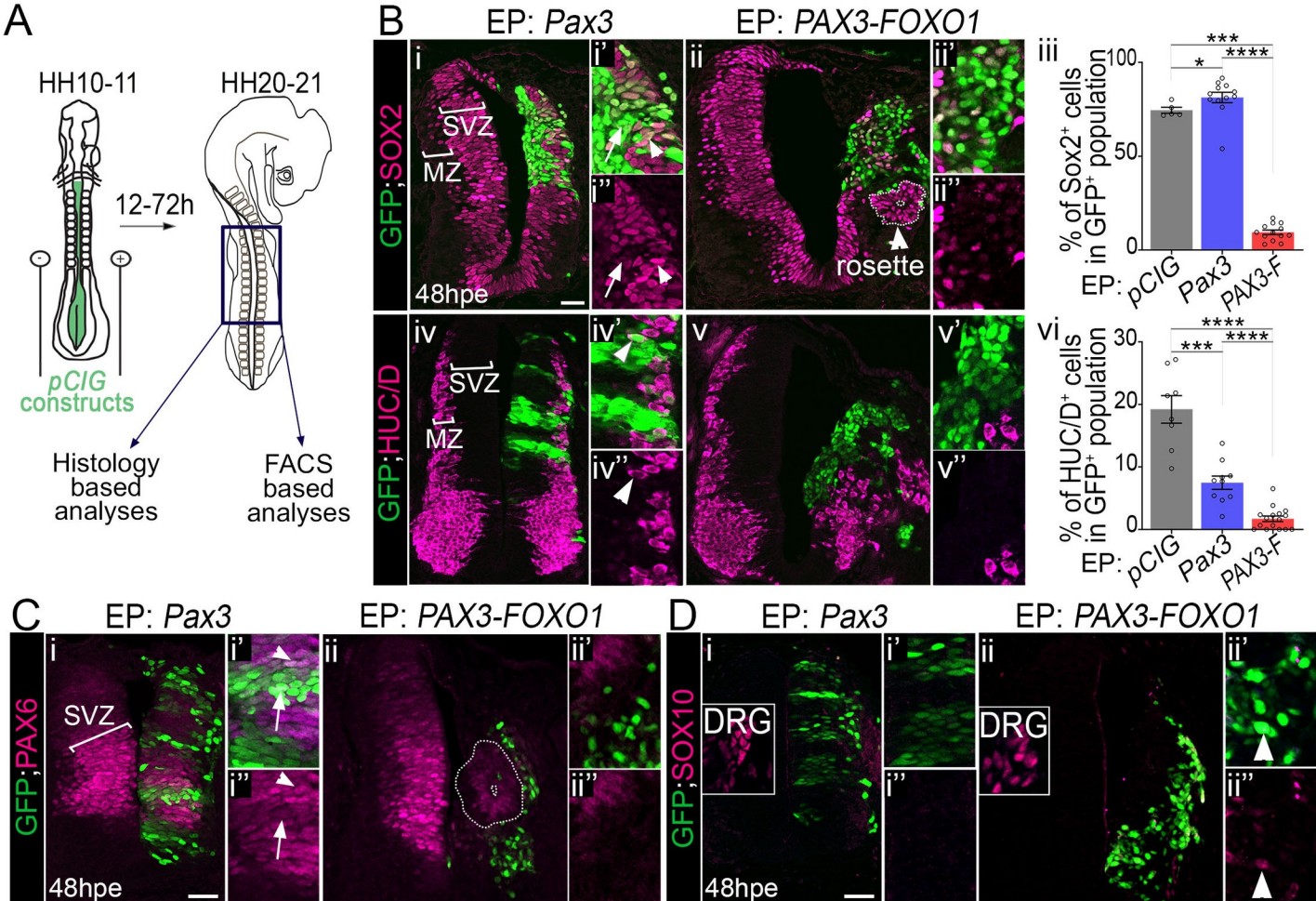

**Fig 1. PAX3-FOXO1 switches off generic neurogenic marker expression in chick embryonic spinal cord. (A)** Schematics showing HH10-11 chick embryos filled with *pCIG* based constructs before being electroporated. The cervical to thoracic region (dark blue square) of the electroporated embryos were dissected 12 to 72 hours later to perform histology or FACS based analyses. **(B) (i-ii"; iv-v")** Immunodetection of GFP, SOX2 and HUC/D on transverse sections of chick embryos 48hpe with the indicated plasmids. MZ: Mantle Zone; SVZ: Sub-Ventricular Zone. **(iii, vi)** Percentage of SOX2 and HUC/D+ cells in the GFP+ population 48hpe with the indicated plasmids (dots: embryo values; bar plots: mean ± s.e.m; Mann-Whitney U test: *: p<0.05, ***: p<0.001, ****: p<0.0001). **(C, D)** Immunodetection of GFP, PAX6 and SOX10 on transverse sections of chick embryos 48hpe with the indicated plasmids. DRG: dorsal root ganglia. Arrowhead in Dii',ii" point to rare SOX10+; GFP+ cells. x' and x" panels are blow-ups of a subset of x panel GFP+ cells. Dash lines in Bii and Cii surround rosettes of GFP- cells clustering apart from the SVZ. Arrowheads in Bi',i" and Ci',i" point at a GFP+ cells expressing SOX2 or PAX6, while those in iv', iv" point at a HUC/D+; GFP+ cell. Arrows in Bi',i" and Ci',i" mark GFPhigh cells expressing low levels of SOX2 or PAX6. hpe: hours post-electroporation; scale bars: 50μm.

We characterised the molecular identity of electroporated cells by assaying the expression of generic neuronal markers (Fig 1B and 1C). At 48 hours post electroporation (hpe), the neural tube of chick embryos contained SOX2 and PAX6[+] progenitors located close to the ventricle and HUC/D[+] neurons laterally in the mantle zone (brackets in the non-electroporated neural tube side in Fig 1Bi,iv and 1Ci). PAX3 overexpression did not affect this organisation and cells kept expressing these neurogenic factors (Fig 1Bi-i", iv-iv" and 1Ci-i"). This is consistent with PAX3 being present in the spinal progenitors located in the dorsal half of the developing neural tube. Yet, in some cells expressing high levels of PAX3, SOX2 and PAX6 expression levels were reduced (arrows in Fig 1Bi" and 1Ci"). More significantly, spinal cells overexpressing PAX3 produced less HUC/D[+] neurons and remained in a SOX2[+] progenitor state (Fig 1Bi-i",iii,iv-iv",vi). This phenotype is reminiscent to that caused by the forced expression of another PAX TF member, PAX6, suggesting that the extinction of PAX is required in neural progenitors for their terminal differentiation [40]. In contrast, PAX3-FOXO1 overexpression caused a marked reorganisation on both the ventricular and mantle regions of the neural tube (Fig 1Bii-ii',v-v" and 1Cii-ii'). Strikingly, most PAX3-FOXO1[+] cells lacked both SOX2 and HUC/D and displayed a strong reduction in PAX6 expression (Fig 1Bii-ii',iii,v-v",vi and 1Cii-ii").

We next checked for the expression of SOX10, a marker of neural crest cells (NCC)[41] (Fig 1D). These cells are specified from the dorsal most part of the neural tube, which they leave to colonize distant embryonic tissues, including the dorsal root ganglia (DRG). At 48hpe SOX10[+] NCC were present in the skin and the DRG (insets in Fig 1Di,ii). The electroporation of Pax3 was not sufficient to induce SOX10 expression (Fig 1Di-i") and only rare PAX3-FOXO1[+] cells were positive for this TF (arrowhead in Fig 1Dii-ii"). This rules out the possibility of a switch of neural cells into NCC upon exposure to the fusion TF. Taken together, these results show that PAX3-FOXO1 is sufficient to divert cells from a generic neurogenic program.

## PAX3-FOXO1 TF converts chick neural cells into FP-RMS like-cells

We next tested whether PAX3-FOXO1 expressing cells adopted the identity of alveolar rhabdomyosarcoma cells. To refine the list of genes that define this identity [9], we combined and re-analysed microarray-based tumour transcriptomes obtained from 99 PAX3-FOXO1 and 34 PAX7-FOXO1 positive FP-RMS patients and 59 patients affected by other RMS subtypes (Fig 2A, S1–S3 Tables, S1A Fig) [42–46]. We identified 1194 genes enriched in FP-RMS biopsies; 40% of which were in the vicinity of previously identified PAX3-FOXO1 bound *cis*-regulatory modules (CRM) [11,12] (Fig 2B). This list of genes largely comprised previously identified PAX3-FOXO1 dependent FP-RMS markers, such as *ALK*, *ARHGAP25*, or *FGFR4* [9,47]. Functional annotation of these genes indicated that they encode for developmental regulators of many embryonic lineages known to be dependent on PAX3 and PAX7 activities (Fig 2C, S4 Table) [48], and not exclusively of the muscle lineage. For instance, during early/late somitogenesis in the caudal part of amniotes, *ALK* is found in the spinal cord neurons and peripheral nervous system [49], *ARHGAP25* is weakly expressed by the neural tube and somite cells (cf. chicken expression database http://geisha.arizona.edu/geisha/) and *FGFR4* marks amongst others the somites [50]. The complexity of FP-RMS signature is likely to stem from the unusual combined expression of master TFs which control the development of distinct lineages in the embryo and/or which act at distinct developmental time points. To illustrate this, we focused on nine TFs, namely *EYA2*, *FOXF1*, *LMO4*, *MEOX1*, *MYOD1*, *PITX2*, *PAX2*, *PRDM12* and *TFAP2β* (Fig 2D). In the myogenic lineage, *MEOX1* is the first to be induced and its activity controls the specification and the segmentation of the epithelialized somites [51]. *LMO4* is

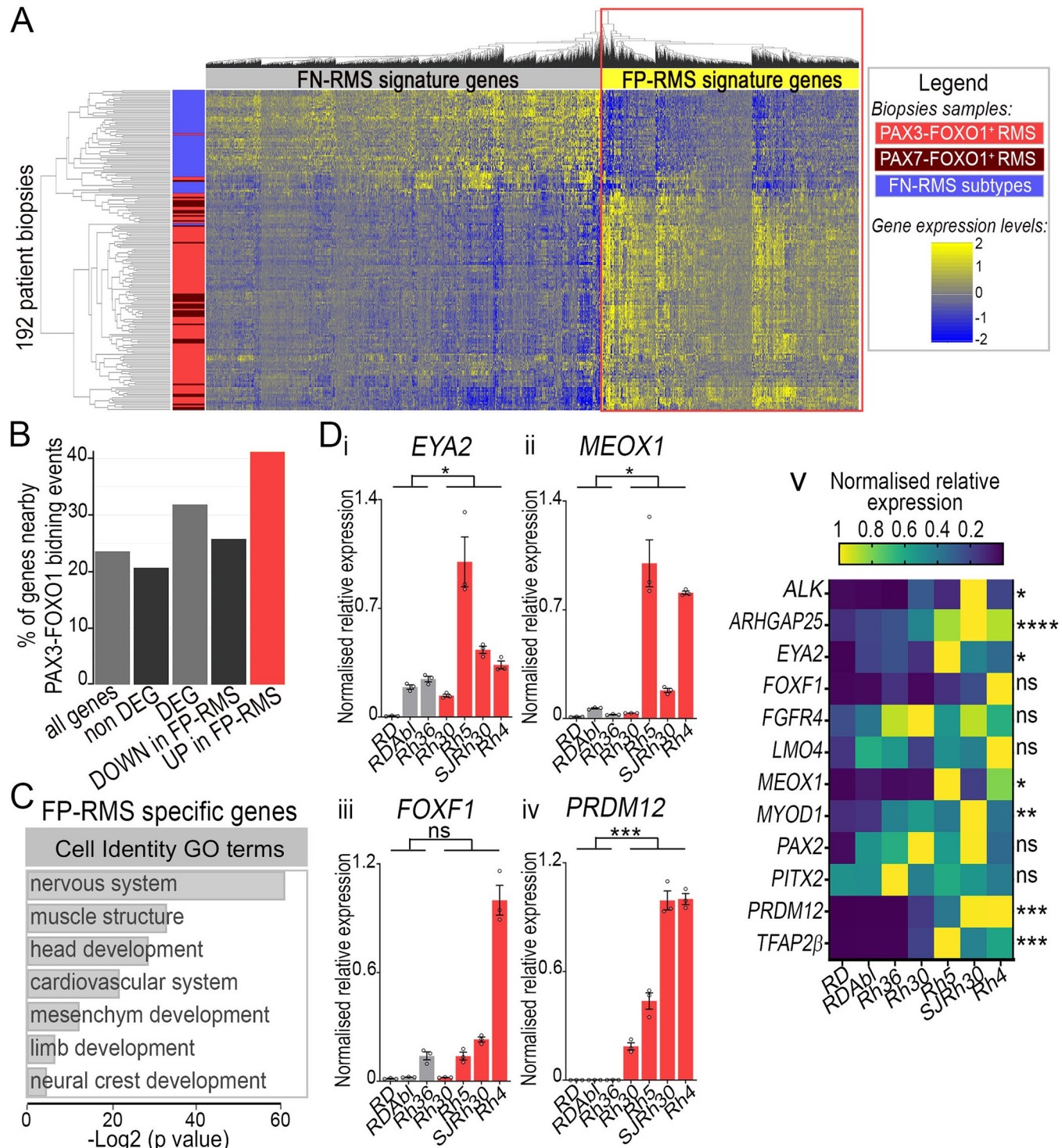

**Fig 2. FP-RMS gene signature is composed of TFs marking in the embryo distinct lineages. (A)** Heatmap of hierarchically clustered differentially expressed genes between PAX3-FOXO1 and PAX7-FOXO1 positive RMS (red and burgundy rectangles, respectively) and FN-RMS biopsies (blue rectangles). Fold changes across samples are colour-coded in blue (lower levels) to yellow (higher levels) (See also Method section and S1–S3 Tables). Genes upregulated in FP-RMS versus FN-RMS are squared in red and named FP-RMS signature genes. **(B)** Percentage of genes present nearby at least one known PAX3-FOXO1 bound CRM [12] out of those present in our complete transcriptomic data set (all genes), non-differentially regulated between FP-RMS and other RMS (non DEG), the differentially expressed genes between FP-RMS and other RMS (DEG), downregulated in FP-RMS compared to other RMS (DOWN in FP-RMS), upregulated in FP-RMS compared to other RMS (UP in FP-RMS). **(C)** Gene ontology enrichment for biological processes terms linked to tissue specification applied to genes enriched in FP-RMS biopsies. **(D)** mRNA expression levels of FP-RMS signature genes nearby a PAX3-FOXO1 binding event and expressed

in various PAX3/7 dependent embryonic tissues assayed by RT-qPCR on the indicated FN-RMS and FP-RMS cell lines. Levels are relative to *TBP* transcripts and normalised to max value across all cell lines. **i-iv:** dots: biological replicates; bar plots: mean ± s.e.m.; n = 3 replicates. **v:** heatmap displays mean value in each cell line. Normalised relative expression across samples are colour-coded in blue (lower levels) to yellow (higher levels). Two-way-ANOVA p-values evaluating the similarities between FP-RMS and FN-RMS cells lines: *: p<0.05, **: p< 0.01, ***: p<0.001, ****: p<0.0001, ns: p>0.05.

transiently induced in the somites [52], while *EYA2* remains longer in these structures where it contributes to the induction of one of the core myogenic determinants *MYOD1* [53,54]. *PITX2* has been shown also to contribute to the induction of *MYOD1* but in the limb myoblasts [55], while its activity in the trunk myoblasts is required at foetal stages [56]. Only briefly expressed in the somites, *FOXF1* marks other mesoderm derived tissues, including the splanchnic mesoderm or the sclerotome [57]. The other TFs, *PAX2* [58], *PRDM12* [59], *TFAP2β* [60] are found in neurons of the peripheral and/or central nervous system, so are *EYA2* [61], *LMO4* [62] and *PITX2* [63]. This suggests that FP-RMS cells are not simply undifferentiated muscle cells, but rather as cells with their own transcriptional status. To verify the presence of this combination of TFs in FP-RMS, we quantified their expression levels using either RT-qPCR or western blots across seven established human RMS cell lines, including 3 FN-RMS (RD, RDAbl, Rh36) and 4 PAX3-FOXO1 positive RMS cell lines (Rh30, SJRh30, Rh4, Rh5) (Fig 2D, S1B Fig, S1 Methods, S1–S4 Raw images). All markers assessed were present in FP-RMS cell lines, with transcript and protein expression levels varying from one cell line to another (Fig 2D, S1B Fig). *EYA2*, *MEOX1*, *MYOD1*, *PRDM12* and *TFAP2β* displayed significant elevated levels in FP-RMS cells compared to FN-RMS cells (Fig 2Di,ii,iv,v). Yet, MYOD1 protein levels did not discriminate FN-RMS and FP-RMS cell lines (S1Biii Fig). *FOXF1* was very high in the FP-RMS Rh4 cells (Fig 2Diii,v). *LMO4* and *PITX2* transcripts were detected in all RMS subtypes (Fig 2Dv). Instead, PITX2 protein isoforms were higher in FP-RMS Rh5, SJRh30 and Rh4, cell lines than in the other cell lines (S1Biii Fig), representing a potential FP-RMS biomarker and supporting post-translation regulation. Altogether these results further highlight inter-patient heterogeneity in the combination of TFs expressed by FP-RMS [64], which could in turn underpin the transcriptomic heterogeneity (Fig 2A). Importantly, they confirmed that the nine TFs chosen can be used to define a FP-RMS identity and to discriminate this identity from other embryonic lineage, notably the myogenic one.

We next assessed the expression of these nine TFs and that of the FP-RMS hallmark genes, *ALK*, *ARHGAP25* and *FGFR4*, in *GFP*, *Pax3* or *PAX3-FOXO1* electroporated chick neural cells (Fig 3A). For this, the neural tube of 48hpe embryos were dissected, dissociated and FACS purified (Fig 1A). RNA from 60 to 80k GFP positive cells was extracted and RT-qPCR used to assay specific cDNAs. The expression of all genes was significantly increased by PAX3-FOXO1 and barely altered by PAX3 (Fig 3A). *In situ* hybridization for *PITX2*, *LMO4* and *MYOD1* performed 24 hours earlier confirmed the ectopic induction of these genes by PAX3-FOXO1 and the absence of these genes in the neural tube submitted to PAX3 ectopic expression (Fig 3B). PAX3-FOXO1 mediated *PITX2* and *LMO4* induction was observed in all the electroporated cells (Fig 3Bii-ii', iv-iv'). Instead, the induction of *MYOD1* by the fusion factor was seen in only half of the electroporated cells (Fig 3Bvi,vi'). In addition, quantification of TFAP2α and PAX2 protein levels after fluorescent immunolabelling of 48hpe embryos showed that PAX3-FOXO1 promotes their expression (Fig 3Cii-ii",iii, S2Aii-iii Fig). Conversely, forced expression of PAX3 had no effect on TFAP2α+ neurons, but inhibited the formation of PAX2+ neurons (Fig 3Ci-i",iii, S2Ai,i',iii Fig). Altogether our data provides evidence that PAX-FOXO1 factors have the ability to induce a molecular signature reminiscent of human FP-RMS cells in neural cells, a non-muscle lineage. It is worth noting that although MYOD1 was induced by PAX3-FOXO1, another member of the core myogenic transcriptional network, namely,

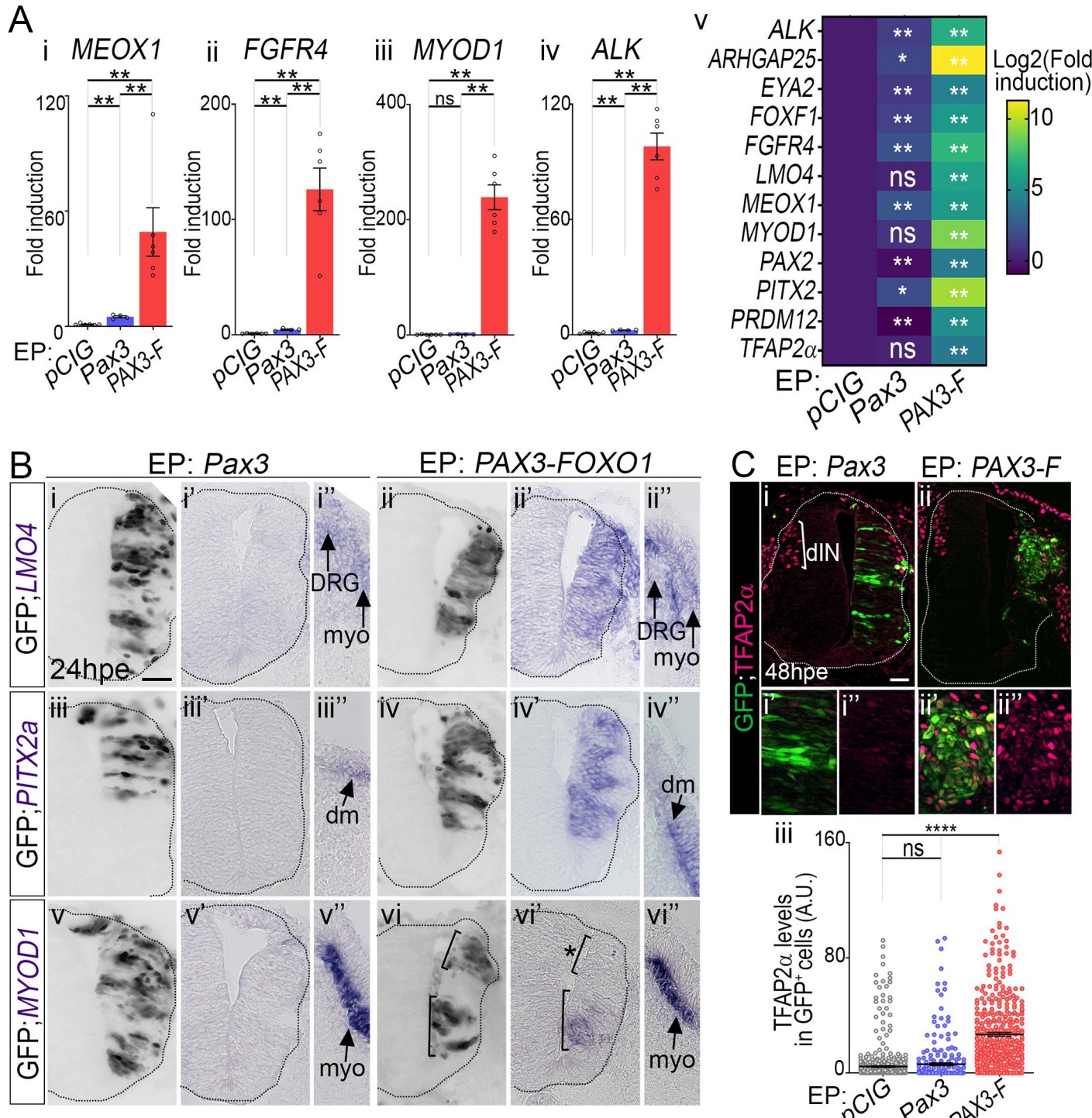

**Fig 3. PAX3-FOXO1 converts embryonic neural progenitors into cells harbouring FP-RMS molecular traits. (A)** mRNA expression levels of FP-RMS hallmark genes in GFP+ FACS sorted neural tube cells 48hpe with the indicated plasmids. Levels are relative to *TBP* transcripts and normalised to *pCIG* samples mean level. **i-iv:** dots: FACS RNA samples; bar plots: mean ± s.e.m; n> 4 FACS sorts; **v:** heatmap exhibits mean value over 4 discrete FACS sorts. Fold induction across samples are colour-coded in blue (lower levels) to yellow (higher levels). **(B)** *In situ* hybridization for *LMO4*, *PITX2a* and *MYOD1* detection and immuno-detection of GFP on transverse sections of chick embryos 24hpe with *Pax3* and *PAX3-FOXO1*. x and x' panels represent the same neural tube but in adjacent histological slides. x" panels display the somitic region of the x' sample. dm: dermo-myotome; DRG: dorsal root ganglia, myo: myotome. Upper bracket in vi, vi' marks cells negative for *MYOD1* (*), the lower one cells positive for this TF. *PITX2a*, *LMO4*: n>9 embryos; *MYOD1*: n = 4 embryos. **(C) (i-ii")** Immunodetection of GFP and TFAP2α on transverse sections of chick embryos 48hpe with the indicated plasmids. **(iii)** Quantification of expression levels of TFAP2α in GFP+ cells in the spinal cords of chick embryos

48hpe with the indicated plasmids (dots: cell values; bars: mean ± s.e.m; n>5 embryos). Mann-Whitney U test p-values evaluating the similarities between either *pCIG* and *Pax3* samples or *pCIG* and *PAX3-FOXO1* samples: *: p<0.05, **: p< 0.01, ***: p<0.001, ****: p<0.0001, ns: p>0.05; Scale bars: 50μm.

MYOG, recurrently used as a RMS marker [64], was not induced by the fusion TF, nor by PAX3 (S2B Fig).

### PAX3-FOXO1 activates conserved FP-RMS associated enhancers in chick neural cells

The robustness of PAX3-FOXO1 mediated FP-RMS hallmark gene induction in neural cells could stem from the activation of conserved enhancers known to operate in FP-RMS cells [11,12]. To test this idea, we cloned enhancers found in the vicinity of the mouse *Met*, *Meox1*, *Myod1*, *Alk*, or human *CDH3* and *PRDM12* genes (S1 Methods). We cloned these upstream of a minimal promoter and a reporter gene and co-electroporated them with either *pCIG*, *Pax3*, *or PAX3-FOXO1*. In the neural tube of embryos electroporated with the control vector the activity of these enhancers was barely detectable (Fig 4i,i',iv,iv'), with the exception of one CRM near the *PRDM12* locus that had an endogenous activity in the intermediate-dorsal neural tube (S2Ci,i' Fig). In contrast, in presence of PAX3-FOXO1 all cloned enhancers, except the *CDH3* CRM, were transcriptionally active and induced a robust and high expression of the reporter gene (Fig 4iii,iii',vi,vi',vii, S2Ciii,iii',iv and S2D Fig). The magnitude of PAX3-FOXO1 mediated induction varied between enhancers and from cell to cell. In contrast, PAX3 transcriptional potential was dependent on enhancers (Fig 4ii,ii',v,v',vii, S2Cii,ii',iv Fig). It promoted *Meox1* CRM activity to levels found in PAX3-FOXO1⁺ cells, while the induction of *Myod1* and *PRDM12* CRM activity by PAX3 was milder than by PAX3-FOXO1. Finally, PAX3 did not promote *Met* nor *Alk* CRMs activity. Altogether these results support a model whereby the transformation of neural progenitors to an FP-RMS cell identity could be mediated by PAX3-FOXO1 co-option of conserved enhancer elements.

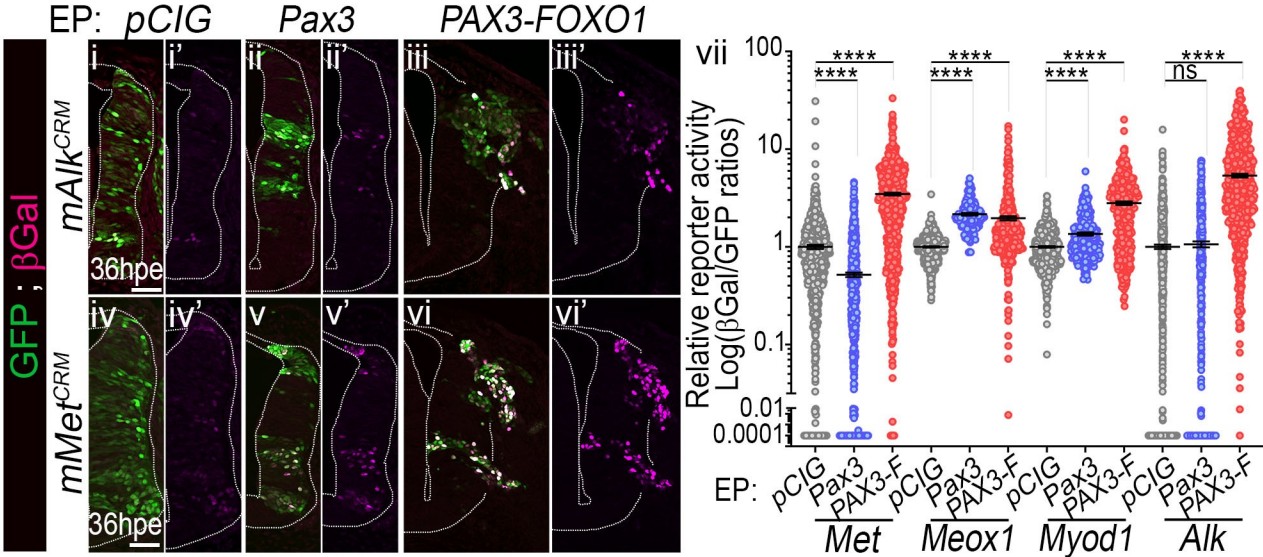

**Fig 4. Activation of FP-RMS associated enhancers in chick neural cells by PAX3-FOXO1. (i-vi')** Immunostaining for GFP and βGalactosidase (βGal) on transverse sections of chick embryos 36hpe with *pCIG*, *Pax3* or *PAX3-FOXO1* and the indicated reporters for the mouse versions of *cis*-regulatory modules (CRM) bound by PAX3-FOXO1 in FP-RMS cells [12]. **(vii)** Quantification of βGal levels normalised to that of GFP in cells electroporated with the indicated enhancer reporter constructs at 36hpe (dots: cell values; bars: mean ± s.e.m.; n>4 embryos). Mann-Whitney U test p-values: ****: p< 0.0001, ns: p> 0.05. Scale bars: 50μm.

## PAX3-FOXO1 promotes epithelial-mesenchymal transition, cell migration and tissue invasion

Paralleling PAX3-FOXO1 mediated cell fate changes, drastic rearrangement of the pseudo-stratified neuro-epithelium occurred (compare Fig 5Aiii,iii' to Fig 5Ai,i'). PAX3-FOXO1$^+$ cells adopted a rounded shape, were unevenly distributed within the tissue, yet grouped together (Fig 5Aiii,iii'). Some cells had delaminated either inside the neural tube canal or within the adjacent mesodermal tissue (brackets in Fig 5Aiii,iii'). In addition, neighbouring unelectroporated cells clustered together to form SOX2$^+$ rosettes ectopically positioned within the "mantle zone", supporting a sorting of PAX-FOXO1$^+$ cells from the non-electroporated ones (circled with dash-lines in Figs 1Bii, 1Cii and 5Aiii). In contrast, cells electroporated with *pCIG* or *Pax3* were aligned, elongated and confined to the neural tube (Fig 5Aii-ii'). In addition, PAX3 overexpression resulted in a thinner neuro-epithelium than seen in *pCIG* samples (compare Fig 5Ai-ii' to Fig 5Ai,i').

To validate these observations, we quantified several parameters in whole embryos stained with the DNA dye DRAQ5 and GFP and documented the distribution of several key markers of the epithelial state (S3A Fig, S6 Table). The tumour cell modes of migration are tightly connected to cell shape (e.g. [65]). Hence, we started by evaluating that of GFP$^+$ cells by measuring the ellipticity of their nuclei segmented from 3D images (S3B Fig). This parameter reflects the degree of divergence from a sphere. It fluctuated between 0.4 and 0.42 for *pCIG* and *Pax3* elongated nuclei (S3Bi,ii,iv Fig). The ellipticity of PAX3-FOXO1$^+$ cells was substantially smaller; with time this difference was accentuated (S3Biii,iv Fig). PAX3-FOXO1 cells adopt a rounded shape likely adapted to tissue exploration [65].

We then monitored the orientation of the major axis of the ellipsoid fit of GFP$^+$ cells using polar coordinates, a parameter indicative of cell arrangement within the tissue (S3C Fig). The polar angle $\theta$ gave the deviation of the nuclei major axis to the tissue dorso-ventral axis, while the azimuthal angle $\varphi$ informed on its orientation within the lateral-medial and posterior-anterior tissue plane (S3Ci Fig). In 48hpe controls and *Pax3* samples, the distribution of $\theta$ and $\varphi$ was similar (S3Cii,iii Fig). $\theta$ was centred on 90˚C, $\varphi$ on 0˚C, consistent with nuclei paralleling the medial-lateral axis of the embryos and apico-basal attachments of cells. In contrast, in *PAX3-FOXO1* samples $\theta$ and $\varphi$ values displayed a wide distribution (S3Cii,iii Fig), ranging for instance for $\varphi$ between—90˚ and + 90˚. Hence, PAX3-FOXO1 is able to randomize the nuclei orientation within the spinal tissue.

Alterations in the shape and orientation of the nuclei by PAX3-FOXO1 led us to assess the apical-basal attachment of cells (Fig 5B, S4A Fig) [66]. We monitored the distribution of the apical determinant PARD3. In open book preparations of 48hpe whole spinal cord, PARD3 labelling revealed a honeycomb-like network at the apical surface (Fig 5Bi). This network remained intact upon gain for PAX3 although cells harboured less cell-cell contacts (Fig 5Bii). In contrast, PAX3-FOXO1 completely disassembled this network (Fig 5Biii,iv). The loss of apical polarity upon PAX3-FOXO1 forced expression was confirmed by looking at the apically located activated form of βCATENIN (S4Ai-ii" Fig). We next looked at the distribution of the focal adhesion anchor β1 INTEGRIN, which accumulates within the basal regions of control cells (arrowheads in S4Aiii-iv' Fig). Upon PAX3 over-expression, the expression levels of this protein increased (arrows in S4Aiii' Fig), yet higher levels of β1 INTEGRIN were detected in the basal region of cells (arrowheads in S4Aiii' Fig). In contrast, the expression of this protein was homogenous throughout PAX3-FOXO1$^+$ cells (S4Aiv' Fig). Hence, upon PAX-FOXO1 expression, neural progenitors lose the polarized distribution of cell-cell and cell-matrix attachments that become distributed evenly throughout their cell membrane.

Because cell polarity is influenced by the extra cellular matrix (ECM) [66], we investigate the distribution of LAMININ (Fig 5Bv-vi'). This key scaffold component of the basal lamina

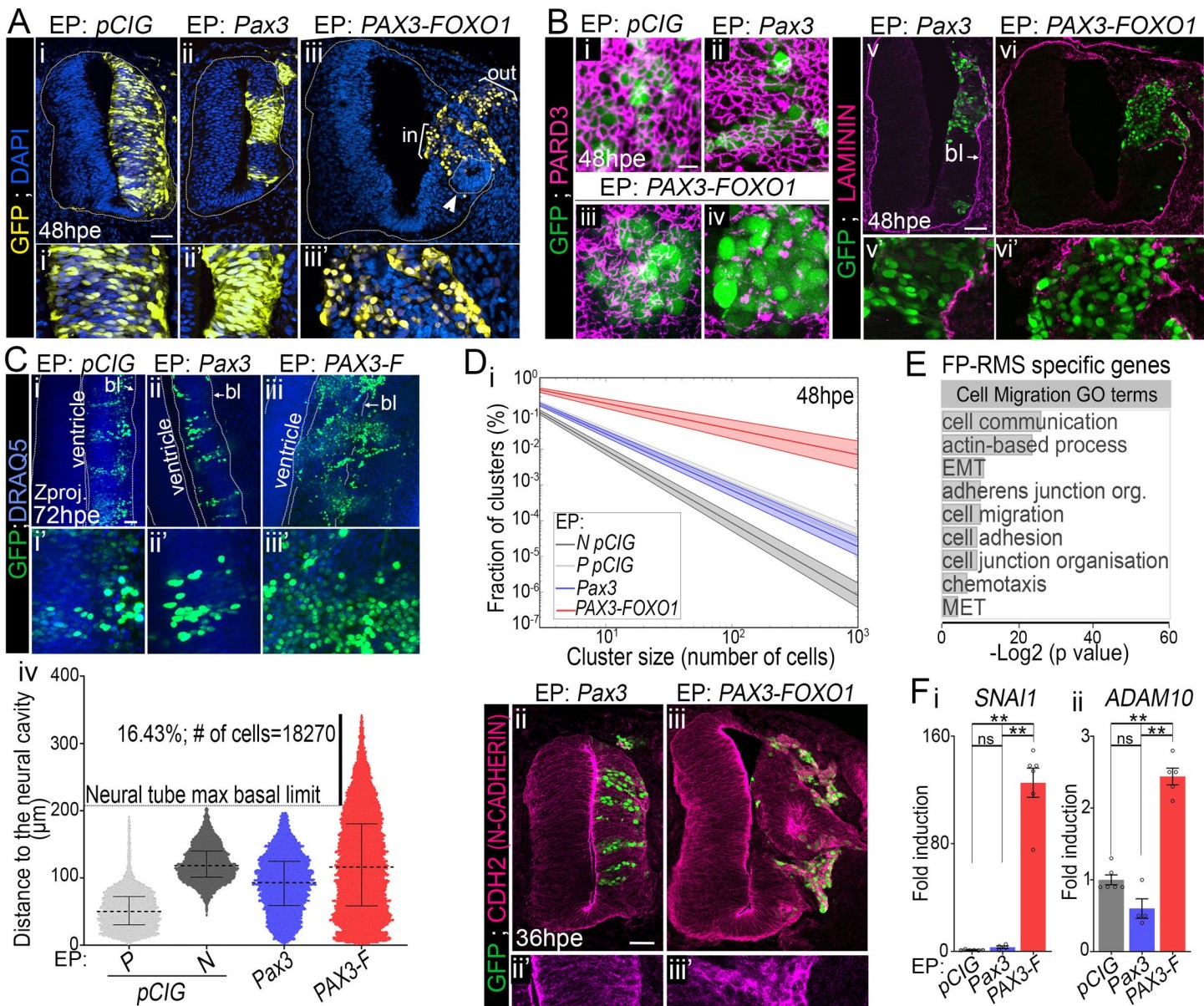

**Fig 5. PAX3-FOXO1 transforms neural epithelial cells into a cohesive mesenchyme capable of migration. (A) (i-iii')** Immunodetection of GFP and DAPI staining on transverse section of chick embryos 48hpe with the indicated plasmids. Brackets in Aiii highlight cells delaminating inside the neural tube (in) or outside of the neural tissue (out). **(B) (i-iv)** Apical confocal views in open-booked preparation of spinal cords of embryos 48hpe with the indicated constructs and immunolabelled with antibodies against GFP and PARD3. Variations in the phenotype are observed in presence of PAX3-FOXO1: (iii) represents 2 of 8 cases analysed, (iv) the rest of cases. **(v-vi')** Immunodetection of GFP and LAMININ on transverse sections of chick embryos 48hpe with the indicated plasmids. **(C) (i-iii')** Z-projections of whole mount 72hpe embryos immuno-labelled for GFP and stained with DRAQ5. Dotted lines delineate either the neural tube ventricle or the neural tube/mesoderm border. **(iv)** Quantification of the distance of each GFP$^+$ nuclei from the apical surface at 72hpe with the indicated plasmids (Violin plots) P: progenitors and N: neurons. **(D) (i)** Exponential fit of the % of cell clusters as the function of cluster size at 48hpe in discrete sample types. **(ii-iii')** Immunodetection of GFP and N-CADHERIN on transverse sections of chick embryos 48hpe with the indicated plasmids. **(E)** Gene ontology enrichment for biological processes terms linked cell migration and adhesion applied to genes enriched in FP-RMS biopsies. EMT: epithelial to mesenchymal transition; MET: mesenchymal to epithelial transition. **(F) (i-ii)** Levels of mRNA expression of the indicated FP-RMS signature gene assayed by RT-qPCR on GFP$^+$ FACS sorted neural tube cells 48hpe with *pCIG*, *Pax3* and *PAX3-FOXO1*. Levels are relative to *TBP* transcripts and normalised to *pCIG* samples mean level (dots: value for a single RNA sample; mean ± s.d.; Mann-Whitney U test p-value: *: p<0.05, **: p<0.01, ns: p>0.05). x' and x" panels are blow-ups of a subset of x panel GFP$^+$ cells. bl: basal lamina. Scale bars: 50µm, but in D: 10µm.

separates the neural tube from the adjacent mesoderm, as shown in *Pax3* samples at 48hpe (Fig 5Bv,v'). In the presence of PAX3-FOXO1, the basal lamina broke down (Fig 5Bvi,vi'). PAX3-FOXO1 provides, thus, cells with the ability to dismantle tissue barriers. We next tested whether PAX3-FOXO1$^{+}$ cells were capable of migration, by measuring the distances between the centre of electroporated nuclei and the apical surface of the neural tube in the 3 dimensions of 72hpe embryos (Fig 5C). In *pCIG* samples, the arrangement of progenitors and neurons nuclei differed and allowed to distinguish these two types of cells. This could not be done in *Pax3* samples, probably because too few neurons were produced and in PAX3-FOXO1 due to the global transformation of cells (Fig 1B). While, nuclei overexpressing PAX3 remained in the neural tube (compare Fig 5Cii,ii' to Fig 5Ci,i' and Fig 5Civ), a fraction of PAX3-FOXO1$^{+}$ cells (more than 15%) had migrated outside the neural tube and were present within the adjacent tissues (Fig 5Ciii-iii', iv).

To investigate whether cells clustered together, we first measured the distance between nearest nuclei from which we evaluated the number of cells belonging to the same cluster (Fig 5Di, see Methods). In control embryos, electroporated neuro-progenitors were more clustered together than neurons, which is in agreement with the delamination and various migration paths taken by neuronal subpopulations (Fig 5Di). PAX3 electroporated cells behave similarly to control neural progenitors (Fig 5Di), as expected from the progenitor like state adopted by PAX3 electroporated spinal cells (Fig 1B). By contrast, in PAX3-FOXO1 electroporated neural tubes, we identified more cells close to each other and bigger groups of cells than in control (Fig 1Di), supporting the idea that it favours the clustering of cells. In agreement with this, PAX3-FOXO1$^{+}$ cells expressed high levels of N-CADHERIN (CDH2), which was homogenously distributed throughout the cells (Fig 5Diii,iii'), while the gain of PAX3 barely modified the apical-basal distribution of CDH2 (Fig 5Dii,ii').

Taken together, these data indicate that PAX3-FOXO1 not only triggers acquisition of FP-RMS identity markers but also provide cells with the ability to invade tissues. These phenotypes are likely to be directly regulated by the fusion TF, as suggested by the great number of PAX-FOXO1 targets in FP-RMS cells encoding for tissues remodellers and cell migration regulators (Fig 5E, S4B Fig). We notably confirmed that the master epithelial-mesenchymal transition driver *SNAI1* and the ECM remodeller *ADAM10* genes displayed elevated levels in presence of PAX3-FOXO1 compared to control and PAX3$^{+}$ chick neural cells (Fig 5F).

## PAX3-FOXO1 holds cells in G1 by decreasing CDK-CYCLIN activity

We next assessed the impact of PAX3-FOXO1 on other hallmarks of cancer cells, notably those related to their cycling behaviour [14](Fig 6, S5 and S6 Figs).

To assess the proliferative state of cells, we marked mitotic cells using an antibody against the phosphorylated form of histone H3 (PH3) (Fig 6A). This indicated PAX3-FOXO1$^{+}$ cells displayed a lower rate of mitosis than control cells at 48hpe (Fig 6Aii-ii",iii). Such a reduction in the number of PH3$^{+}$ cells was seen also in PAX3$^{+}$ cells, albeit to a lesser extent (Fig 6Ai-i", iii). These results suggest that either PAX3-FOXO1$^{+}$ cells were blocked in a cell cycle phase or had a longer cell cycle(s). We first evaluated if the fusion protein specifically induced cell death by marking activated CASPASE3$^{+}$ apoptotic cells (S5A Fig). Upon overexpression of PAX3 or PAX3-FOXO1, a too low proportion of cells (about 2%) were undergoing cell death at 48hpe to account for the mitotic rate decrease. We next traced cells undergoing DNA synthesis by treating embryos with EdU for 20h before harvesting (Fig 6B). Nearly all control cells were positive for EdU, while only half of PAX3-FOXO1$^{+}$ cells (Fig 6Bii-ii", iii) and 75% of PAX3$^{+}$ cells incorporated the thymidine analogue (Fig 6Bi-i',iii). Confirming this compromised entry into replication, the expression levels of minichromosome maintenance 2 (MCM2), a protein

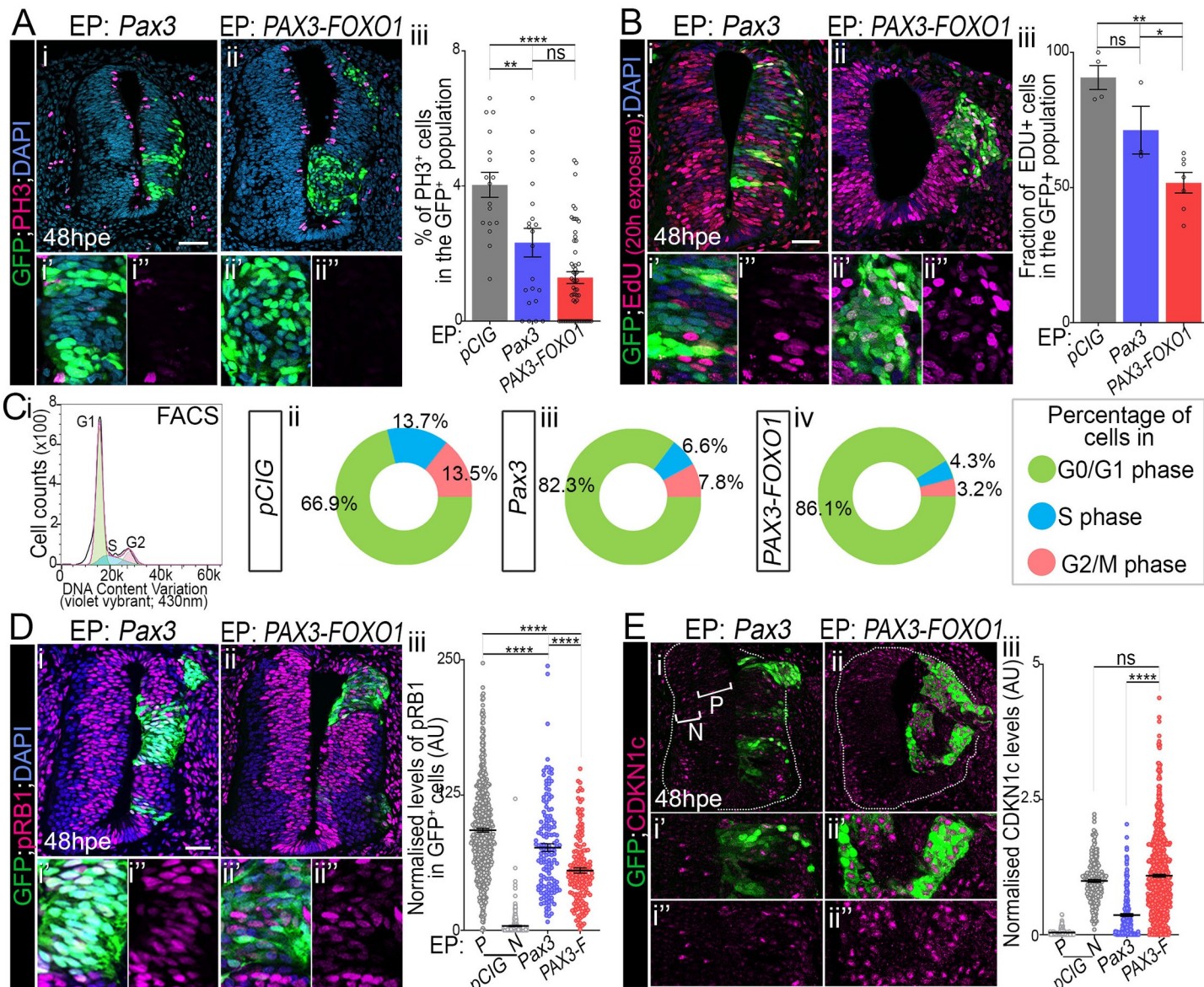

**Fig 6. Pax3 and PAX3-FOXO1 limit the entry of cells into S phase. (A) (i-ii')** Immunodetection of GFP, phospho-histone H3 (PH3) and DAPI staining on transverse sections of chick embryos 48hpe with the indicated plasmids. **(iii)** Quantification of the number of PH3+ cells in the GFP+ population in embryos expressing the indicated plasmids (dots: embryo values; bar plots: mean ± s.e.m.). **(B) (i-ii')** Immunodetection of GFP, EdU and DAPI staining on transverse sections of chick embryos 48hpe with the indicated plasmids and soaked with EDU 20h before harvest. **(iii)** Quantification of the number of EdU+ cells in the GFP+ population in embryos expressing the indicated plasmids (dots: embryo values; bar plots: mean ± s.e.m.). **(C)(i)** FACS plots showing DNA content distribution of GFP+ chick neural cells stained with vybrant dyecycle violet stain and cell cycle phases gating (green: G0/G1 phase, blue: S phase and pink: G2/M phase). **(ii-iv)** Percentage of cells in the indicated cell cycle phase at 48hpe established as in i (mean over three experiments, for individual values see S6Bi-iii Fig and for raw plots see S6A Fig). **(D) (i-ii')** Immunodetection of GFP, phosphorylated form of RB1 (pRB1) and DAPI staining on transverse sections of chick embryos 48hpe with the indicated plasmids. **(iii)** Quantification of pRB1 levels in the GFP+ cells in embryos expressing the indicated plasmids (dots: cell values in arbitrary unit (AU), bars: mean ± s.e.m., n> 8 embryos). **(E) (i-ii')** Immunodetection of GFP and CDKN1c on transverse sections of chick embryos 48hpe with the indicated plasmids. P: progenitors; N: neurons. **(iii)** Quantification of CDKN1c levels in GFP+ cells in embryos expressing the indicated plasmids. P: progenitors; N: neurons. (dots: cell values, bars: mean ± s.e.m., n>4 embryos). x' and x" panels are blow-ups of a subset of x panel GFP+ cells. Mann-Whitney U test p-value: *: p< 0.05, **: p< 0.01, ***: p< 0.001, ****: p< 0.0001, ns: p>0.05. Scale bars: 50μm.

of the pre-replicative complex was significantly downregulated in PAX3-FOXO1 cells (compare S5Biii-iii" Fig to S5Bi-i" Fig, S5Biv Fig). The levels of this protein were barely affected by the gain in PAX3 expression (S5Bii-ii",iv Fig). FACS analyses after labelling cells with a

permeable DNA dye (vybrant dyecycle violet stain) (Fig 6Ci, S6A Fig), indicated that in presence of PAX3 and PAX3-FOXO1 a larger proportion of cells were in the G1 phase (compare Fig 6Ciii and 6Civ, respectively to Fig 6Cii, S6Bi-iii Fig). Taken together, these results support the idea that the gain for PAX3 or PAX3-FOXO1 arrests cells in G1 phase. Similar experiments performed in human fibroblasts indicated that this cell type was also arrested in G1 upon gain for PAX3-FOXO1 (S6Biv-vi Fig), supporting the idea that PAX3-FOXO1 mediated cell cycle hold is not inherent to our chick system.

Finally, the phosphorylation of the retinoblastoma-associated RB1 protein being one of the hallmarks of the CDK-CYCLIN activity leading to the entry in S phase, we assayed its status (Fig 6D). Both PAX3 and PAX3-FOXO1 decreased phospho-RB1 levels; the fusion protein to a greater extent than wild-type PAX3 (Fig 6D). Yet, it is worth noting that phospho-RB1 was still detected in the cells overexpressing the PAX variants and were higher than cells that have left the cell cycle, such as the neurons (Fig 6Diii). Hence, cells are probably not fully arrested. The decrease in phospho-RB1 levels is not linked to a decrease in the transcription of *RB1*, *CDK2*, *CDK6* and *CCND1* (S5C Fig). Instead, we identified that amongst the CIP/KIP CDK inhibitors, CDKN1c (P57Kip2) was upregulated by the fusion protein (Fig 6E), a cue potentially explaining the PAX3-FOXO1 mediated decrease in CDK-CYCLIN activity.

## PAX3-FOXO1 mediated cell cycle inhibition is overcome by CCND1 or MYCN

We then wanted to test whether PAX3-FOXO1-transformed cells could re-enter cell cycle. For this, we first decided to reactivate CDK-CYCLIN activity in PAX3-FOXO1 expressing cells, by forced expression of CYCLIN D1, CCND1 [67]. In presence of this cyclin subtype, PAX3 and PAX3-FOXO1 positive cells displayed a mitotic rate, revealed by quantifying PH3$^+$ cells, similar to that of *pCIG* control embryos at 48hpe (compare Fig 7Ai-ii" to Fig 6Ai-ii", Fig 7Aiii). Accordingly, the gain for CYCLIN D1 allowed PAX3-FOXO1$^+$ cells to incorporate EdU as do controlled cells (compare Fig 7Bi-i" to Fig 6Bii-ii", Fig 7Bii).

We next wondered whether the proto-oncogenes recurrently amplified in FP-RMS cells could also overcome the G1 arrest of PAX3-FOXO1 expressing cells. MYCN been amplified in about 10% of FP-RMS [6], we forced its expression together with PAX3-FOXO1. In the control neural tube, as previously demonstrated [68], MYCN, as opposed to its usual function, decreased the number of progenitors in M phase (Fig 7Aiii). In contrast, in presence of PAX3-FOXO1$^+$ and MYCN cells became more actively proliferative (Figs 7Aiii and S5D), with their rate of mitosis reaching levels comparable to that of control *pCIG* embryos.

Finally, we checked that upon reactivation of the proliferative activity of PAX3-FOXO1, the specific traits induced in the transformed neural progenitors were maintained. Assaying the migration of cells supported this idea (Fig 7C) and expression of the FP-RMS marker gene *PITX2* (Fig 7D).

Together, these results indicated that PAX3-FOXO1 proteins inhibit the entry of cells into S phase and this is associated with a decrease in CDK-CYCLIN activity. This inhibition can be overcome by increasing the levels of CYCLINs or that of MYCN.

## PAX7-FOXO1 transformation of spinal progenitors is reminiscent to that by PAX3-FOXO1

Finally, we assessed whether the transformation properties of PAX3-FOXO1 were shared by PAX7-FOXO1 and how much the effects of PAX7-FOXO1 on spinal progenitors diverge from that of PAX7 (Fig 8, S7 Fig). To do so, we focused on 3 features. First, we assayed cells 48hpe with *PAX7-FOXO1* or *Pax7* using the pan-neuronal markers SOX2 and HUC/D and the

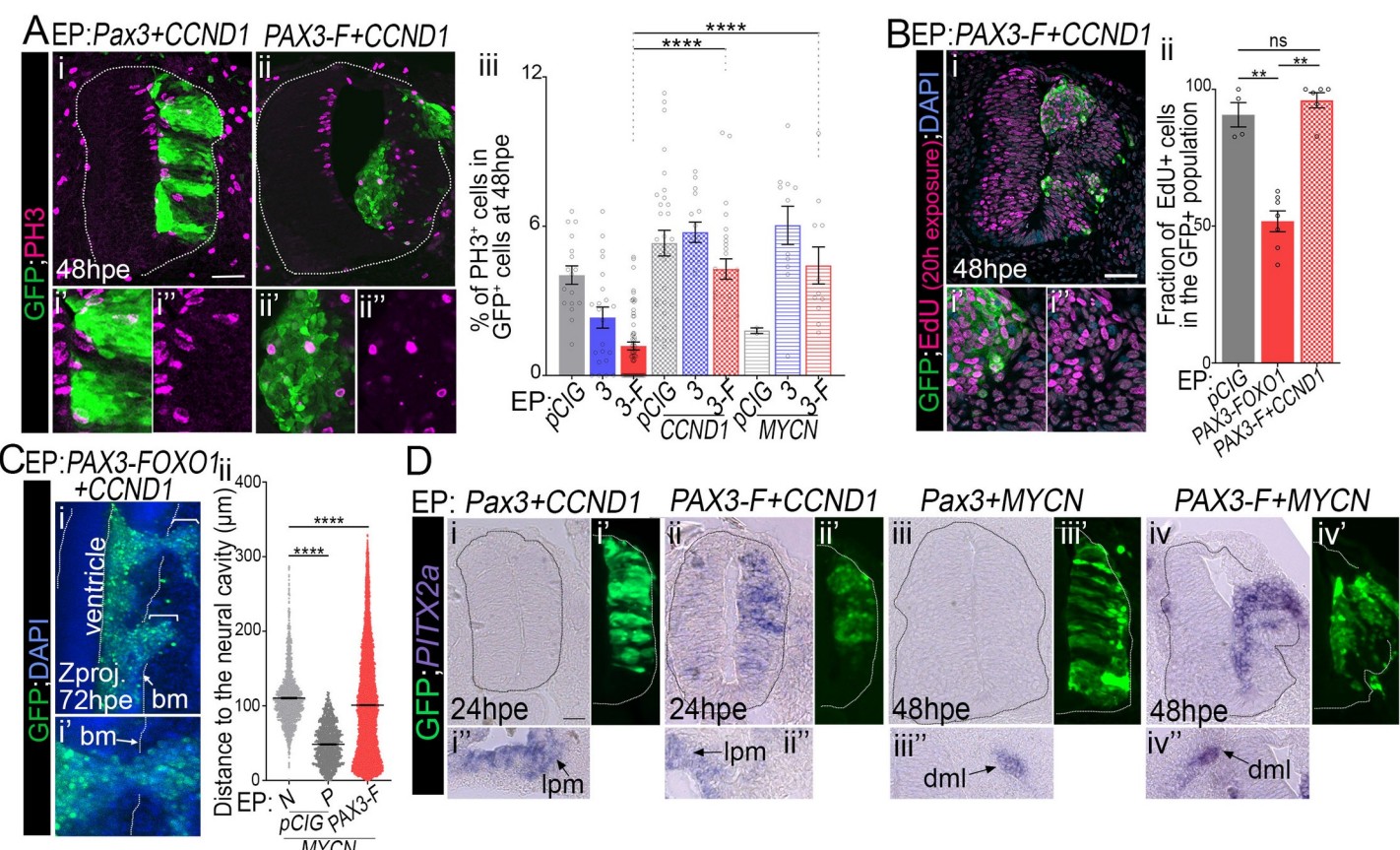

**Fig 7. CYCLIN D1 and MYCN rescue PAX-FOXO1 cell cycle inhibition, without affecting the identity and migration of cells. (A) (i-ii")** Immunodetection of GFP, the phosphorylated form of histone H3 (PH3) and DAPI staining on transverse section of chick embryo at 48hpe with the indicated plasmids. **(iii)** Quantification of the number of PH3+ cells in the GFP+ cells in embryos expressing the indicated plasmids at 48hpe. (dots: embryo values; bar plots: mean ± s.e.m.). Data for *pCIG*, *Pax3*, *PAX3-FOXO1* samples are the same as in Fig 6Aiii. **(B) (i-i")** Immunodetection of GFP, EdU and DAPI staining on transverse section of chick embryos 48hpe with the indicated plasmids and soaked with EDU 20h before harvest. **(ii)** Quantification of the number of EdU+ cells in the GFP+ population in embryos expressing the indicated plasmids (dots: embryo; bar plots: mean ± s.e.m.). **(C) (i-i')** Z projection along the dorso-ventral axis of 3D scans of an embryos 72hpe with *PAX3-FOXO1* and *CCND1*. Dotted lines mark the apical cavity and the basal membrane (bm). **(ii)** Quantification of the distance of each GFP+ nuclei and the apical surface at 72hpe with the indicated plasmids (Violin plots). **(D)** *PITX2* detection via *in situ* hybridization on transverse sections of chick embryos 24hpe and 48hpe with the indicated plasmids and immuno-detection of GFP on the adjacent section slide. Bottom panels show areas on the embryos and section presented in the upper panels where PITX2 is expressed, including the lateral plate mesoderm (lpm) and the dorsal medial lip of the dermomyotome (dml). x' and x" panels are blow-ups of a subset of x panel GFP+ cells. Mann-Whitney U test p-value: **: p< 0.01, ****: p< 0.0001, ns: p> 0.05. Scale bars 50μm.

FP-RMS signature genes *LMO4*, *PITX2a*, TFAP2α, and Pax2 (Fig 8A, S7A and S7B Fig). Forced expression of PAX7, similar to PAX3, promoted the maintenance of a SOX2+ state (S7Ai,i',iii Fig), reduced the formation of HUC/D+ neurons (Fig 8Ai-i",iii) and failed to promote the expression of the selected FP-RMS signature genes (Fig 8Aiv-iv",vi, S7Bi,i',iii,iii',v-v" Fig). By contrast, PAX7-FOXO1 induced FP-RMS markers at the expense of the pan-neuronal markers (Fig 8Aii-iii,v-vi, S7Aii-iii and S7Bii-ii',iv,iv',vi,vi" Fig), as previously shown with PAX3-FOXO1. The levels of TFAP2α in PAX7-FOXO1+ cells reached levels similar to those found in PAX3-FOXO1+ cells (compare Fig 8Avi to Fig 3Ciii), while PAX7-FOXO1 poorly induced *PITX2* and *LMO4* compared to PAX3-FOXO1 (compare S7Bii,ii',iv,iv' to Fig 3Bii,ii', iv,iv'). This may stem from the differential transcriptional potential between the two TFs [10].

Second, we investigated the tissue remodelling properties of PAX7-FOXO1 and PAX7 (Fig 8B). While the latter had little effect on neuro-epithelium structure (Fig 8i,iv), PAX7-FOXO1 triggered marked tissue remodelling (Fig 8Aii,v). As observed for PAX3-FOXO1 gain of

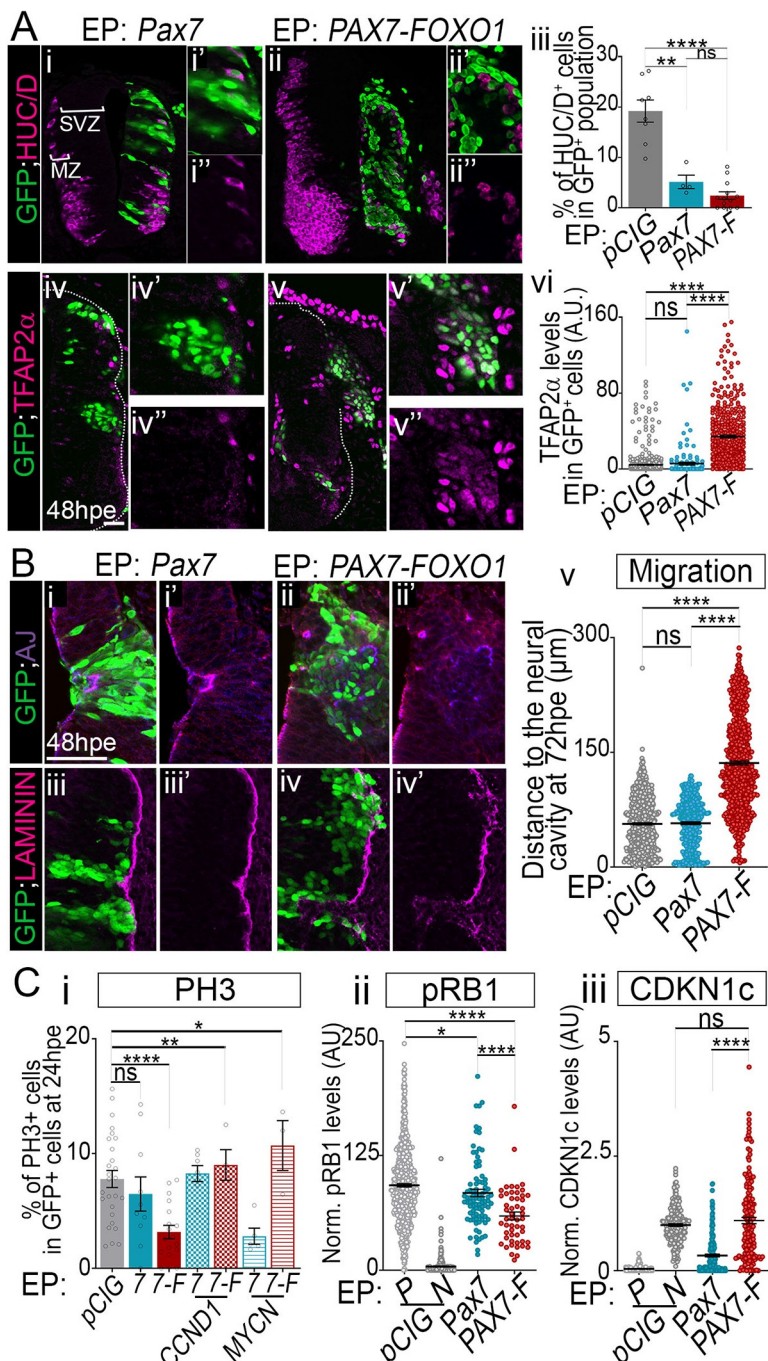

**Fig 8. Major traits of PAX7-FOXO1 cellular transformation. (A) (i-ii", iv-v')** Immunodetection of GFP, HUC/D and TFAP2α on transverse section of chick embryo at 48hpe with the indicated plasmids. **(iii, vi)** Quantification of the number of HUC/D[+] and TFAP2α[+] cells in the GFP[+] cells in embryos expressing the indicated plasmids at 48hpe. (dots: embryo values; mean ± s.e.m.). Data for the *pCIG* samples are the same as presented in Figs 1Bvi and 3Ciii). **(B) (i-iv")** Immunodetection of GFP, activated bCATENIN joined with PARD3 (AJ, adherens junctions) and LAMININ on blows-off on GFP[+] cells of transverse sections of chick embryos 48hpe with the indicated plasmids. **(v)** Quantification of the distance of GFP[+] nuclei to the apical surface of the neural tube in embryos 72hpe with the indicated plasmids measured on transverse sections (dots: values in individual cells; bars: mean ± s.e.m.; n>5 embryos). **(C) (i)** Quantification of the number of PH3[+] cells in the GFP[+] cells in embryos expressing the indicated plasmids at 24hpe. (dots: embryo values; mean ± s.e.m.. **(ii,iii)** Quantification of pRB1 (ii) and CDKN1c (iii) levels in the GFP[+] cells in embryos expressing the indicated plasmids (dots: cell values in arbitrary unit (AU), bars: mean ± s.e.m., n> 5 embryos). Data for the *pCIG* samples in C are the same as presented in S5D Fig, Fig 6Diii and Fig 6Eiii. x' and

x" panels are blow-ups of a subset of x panel GFP$^+$ cells. Mann-Whitney U test p-value: $^{**}$: p$<$ 0.01, $^{****}$: p$<$ 0.0001, ns: p$>$ 0.05. Scale bars 50μm.

function, PAX7-FOXO1 remodelling was accompanied by the loss of restricted accumulation of apical markers (Fig 8Bii-ii') and a breakdown of the basal lamina (Fig 8Biv-iv'). The gain for PAX7 did not alter the distribution of these proteins (Fig 8Bi,i',iii,iii'). Furthermore, PAX7-- FOXO1$^+$ cells gained the ability to colonize adjacent tissues, while the forced expression of PAX7 did not alter the position of electroporated cells within the embryo (Fig 8Bv).

Finally, quantifying the number of PH3$^+$ cells and the levels of phospho-RB1 in chick embryos electroporated with PAX7 or PAX7-FOXO1 were consistent with these PAX variants reducing the ability of progenitors to proliferate via a reduction of CDK-cyclin activity (Fig 8Ci,ii, S7C Fig); with the effects of PAX7-FOXO1 much stronger than that of PAX7. The reduced proliferation of PAX7-FOXO1$^+$ cells was correlated with elevated levels of CDKN1c (Fig 8Ciii, S7Dii-ii" Fig). This was not apparent for PAX7$^+$ cells (Fig 8Ciii, S7Di-i" Fig). We next checked whether the proliferative behaviour of PAX7-FOXO1$^+$ cells could be stimulated by co-expressing CCND1 or MYCN (Fig 8Ci). At 24hpe in presence of either cell cycle regulator PAX7-FOXO1$^+$ cells displayed a mitotic rate as great as that of *pCIG* samples (Fig 8Ci).

Altogether these data demonstrate that the gross phenotypic traits provided by PAX3-- FOXO1 and PAX7-FOXO1 are highly reminiscent, despite some differences in the molecular response of cells to the two factors. Assessing the molecular mechanisms underpinning these differences and how they impact the long term phenotype of cells could provide explanations for the variations in the outcome between patients carrying either the t(2;13)(q35;q14) or t (1;13) (p36;q14) translocations[69].

## Discussion

### New markers of FP-RMS and their regulation by the PAX-FOXO1s

As for many cancers, the transcriptional state of FP-RMS varies considerably between patients (Fig 2A). This is exemplified by variations in the profile of expression of the embryonic muscle markers, MYOD1 and MYOG [64] (S1B Fig). By comparing the transcriptome of 192 RMS patients we have been able to revaluate the list of genes marking the identity of this rare cancer. The distinctive feature of the FP-RMS molecular signature is its association with genes not only regulating the development of embryonic muscle cells, but also of other embryonic cell lineages, having in common a PAX3/7 dependent ontology [11,48](Fig 2C). Accordingly, the chromatin landscape of FP-RMS only partially matches that of myoblasts and myotubes [12]. The presence of PAX3-FOXO1 bound CRMs in the vicinity of 40% of these FP-RMS associated genes and of PAX DNA motifs within at least half of these CMRs represents a means by which PAX3/7 dependent developmental gene networks are co-opted by PAX-FOXO1 expressing cells [11,12,70]. In addition, the presence of TFs known to also strongly influence the differentiation of PAX dependent embryonic lineages [51,53–57,59,62,63,71] might also contribute to the FP-RMS dysfunctional transcriptional state [1,12,72].

Variations in the FP-RMS signature genes is likely to stem from the specific genetic aberrations, the time of appearance of the first genetic aberrations, and the cell of origin [5,6]. Our data demonstrate that the PAX-FOXO1s are able to switch on in the neural tissue some of FP-RMS associated TFs, while their expression is normally silenced (Figs 3 and 8A, S2A and S2B Fig). This is likely stemming from a pioneer transcriptional activity [73], demonstrated for PAX3-FOXO1 in human fibroblasts [12]. There, genomic recruitment of the fusion TF operates largely on closed and transcriptional shut down CRM. PAX3-FOXO1 is able to displace

the CRM nucleosomes and to set up an epigenetic landscape associated with active transcription. In agreement, the PAX-FOXO1s were able to activate *de novo* CRMs in the embryonic neural tube (Fig 4, S2C Fig).

In comparison to the PAX-FOXO1s, the wild-type PAX3 or PAX7 are way more sensitive to the cellular context. Exemplifying this, while *MYOD1* can be induced in embryonic stem cells derived myoblasts by PAX3 [74], we were not able to detect MYOD1 transcripts after a gain for PAX3 expression in the neural tube (Fig 2A and 2B). Tissue specific cofactors or co-modulators of the PAX3/7 TFs haven't yet been revealed. Yet, genetic studies suggest a model where cell fate specification relies on mainly on PAX dependent transcriptional activation in the myogenic lineage (e.g. [75–77]) and in great means on PAX dependent transcriptional repression in the neural tube (e.g. [78–80]). As such PAX3 recruitment to the genome of myoblasts is associated with positive chromatin marks [75] and PAX3 loss of function phenotypes in these cells can be largely rescued by the expression of PAX3-FOXO1 [76]. Conversely, in the neural tube it would act at least partially as a transcriptional repressor [80], PAX3-FOXO1 does not rescue Pax3 loss of function phenotypes [80]. Even more it can interfere with the normal function of PAX3 and leads to specific cell fate changes ([29,80,81] and our study). In agreement, PAX3 poorly induced the activity of FP-RMS in the neural tube of chick embryos (Fig 4, S2B Fig). Interestingly, it has been observed that the fusion with FOXO1 inhibits this repressive activity [82] and therefore the PAX3-FOXO1 proteins harbour a higher transcriptional potential, which certainly relies less than PAX3 on the tissue specific co-factors.

The tissue specific activity of PAX3 can be bypassed, as demonstrated by the induction of *MYOD1* in chick neural explants in which PAX3 is provided by RCAS based viral infection for 5 days[83]. This mode of transgenesis exposes cells to high levels and prolonged expression of the transgenes, as opposed to the electroporated and episomally transcribed plasmids transgenes that get progressively diluted by cell division. This is in line with the idea that the levels of PAX TFs are an important parameter for the response of cells. Exemplifying this, spinal progenitors harbouring different levels of PAX activity generates distinct combinations of neuronal subtypes [84] and the loss of one single *PAX3* allele leads to Waardenburg syndrome, a developmental defect in NCC [85].

The use of various promoters to drive the expression of PAX3-FOXO1 in zebrafish or the comparison of PAX3-FOXO1 effects when expressed from one or two Pax3 alleles support the idea that PAX-FOXO1 levels are also instructive and determine the transformation of healthy cells to FP-RMS like cells [28,29,32]. This may explain the discrepancies between the phenotypes we have observed in the chick neural tube and that reported in the neural tube of mice where PAX3-FOXO1 is either expressed from the endogenous Pax3 locus or using Pax3 promoter region [81,84,86]. In these mouse models, NCC migration defects, mis-specification of neuronal identities, the neural epithelium folding alterations and in some cases ectopic muscles formation have been reported. In contrast, the presence of both myoblast and non myoblasts associated TFs within the chick neural cells expressing PAX3-FOXO1 and PAX7-FOXO1 support a routing of cells a FP-RMS like state.

Most importantly, our study, taken together with the study by Kendall *et al.* [29], supports the contention that neural tube cells are a cellular subtype from which FP-RMS can originate. Accordingly, 20–40% of primary tumour masses are found in organs colonized by NCC, such as the orbit, bladder, para-meningeal, head and neck areas ([41,46,87], S1A Fig). Moreover, several clinical studies report the presence of FP-RMS primary growths in a giant naevus and spinal cord, that are unambiguously neural tube derived [36,37]. This idea is further supported by the observation that the regulatory regions in the vicinity of the *PAX3* locus which are not impacted by the t(2;13)(q35;q14) translocations remain active in the neural tube after the translocation [88]. The impact of this cell origin on the manifestation of the disease and how

much it can contribute the FP-RMS heterogeneity remain to be fully clarified, yet it is tempting to speculate that it will modulate tumour formation incidence, location and histology [29,33].

## PAX-FOXO1s mediated cell cycle inhibition limits the expansion of transformed and metastatic cells

In the light of the cellular phenotypes appearing upon exposure to PAX-FOXO1, we propose that these TFs stand as *sensus stricto* oncogenic drivers, whose activity is likely underpinning the timeline of tumour formation. On the one hand, PAX-FOXO1s rapidly provide cells with tissue remodelling and invasion capacities. This is reminiscent of the transformational power of a small group of TFs, named EMT-TFs [89]. Explaining this, PAX-FOXO1 dependent FP-RMS signature is significantly enriched for key regulators of tissue remodelling. It includes notably modulators of RhoGTPases activity, such as ARHGAP25[47]. Indeed, the Rho GTPases are known to regulate cell-cell and cell-ECM interactions, polarity and migration [90], which are all modulated upon PAX-FOXO1 exposure (Fig 4). In addition, PAX-FOXO1 tissue remodelling activity could be reinforced by the recruitment of other EMT driving TFs, such as SNAI1, PRXX1, ETS1/2 [89] (S1–S3 Tables, Fig 4).

On the other hand, our analyses revealed that the oncogenicity of PAX-FOXO1 transformed cells is limited by their low rate of proliferation (Fig 5). Such negative effect of PAX-FOXO1s on cell cycle progression is unlikely to be inherent to our model system. Human myoblasts expressing PAX3-FOXO1 are not to be able to produce colonies within soft agar [30], it takes several weeks of culture for PAX3-FOXO1[+] NIH3T3 cells to generate such colonies [91], and PAX-FOXO1s[+] human fibroblasts are arrested or spend more time in G1 phase, as do chick neural cells (S3E Fig). These results provide insight for why complementation of PAX-FOXO1s with genetic aberrations promoting cell cycle progression, such as the gain of MYCN or CCND1, or loss of p53, RB1 or CDKN2A, can enhance their tumorigenic potential (Fig 3, Fig 6) [28,30,34]. Whether such complementation is required for FP-RMS' evolution and if so how it is achieved is not known for most cases. Alterations including mutations and small indels, copy number deletions and amplifications or structural variations within cell cycle regulators associated with PAX-FOXO1 generating chromosome translocation is only seen in 30% of biopsies [5,6]. This calls for a better understanding of the molecular mechanisms underpinning this cell cycle inhibition. The buffered cell cycle progression induced by PAX-FOXO1 proteins _a slow cycling state_ could underlie the refractory response of FP-RMS cells to drugs such as CDK2 inhibitors [34] and contribute to the elevated resurgence of tumours post-treatment [92], as shown for other cancers [2]. We propose that RB1 activity inhibition is central to PAX-FOXO1 mediated establishment of a dormant state. The decrease in the levels of the phosphorylated form of RB1 post PAX3-FOXO1 gain of function points at a decrease in the level of CDK2 activity and explains the arrest or longer stay in early G0/G1 [93]. This is further supported by the elevated levels of CDKN1c (p57[Kip2]), a protein that binds to and inhibits CDK2 activity [94], and was originally shown to cause cell cycle arrest mostly in G1 phase. This hypothesis is also compatible with a phenotypic rescue by a complementation with CCND1 (Cyclin D1), an efficient driver to S phase [95]. Strongly supporting the idea that RB1 regulation is a nodal point in PAX-FOXO1 mediated cell cycle regulation, its loss of function have been shown to affect the progression, but not the formation of tumours from p53 null cells of the Myf6 embryonic muscle lineage overexpressing PAX3-FOXO1[34].

Finally, amongst the approaches taken to study FP-RMS development and evolution [28–30,32,33,96], our model uniquely recapitulates the invasive and disseminative properties of PAX-FOXO1 expressing cells [92]. As previously demonstrated with human grafted cells [38], we believe that it will particularly suited for studying the modes of dissemination paths of

PAX-FOXO1 transformed cells. Our model will also provide a means to investigate the molecular networks acting during the transition from a PAX-FOXO1 mediated-latent metastatic state to overt metastasis [97]; and thereby to provide valuable insights for future therapeutics development.

## Methods

### Bioinformatics

Transcriptomes of FP-RMS and ERMS biopsies have been published elsewhere [42–46] (accession numbers GSE92689, E-TABM-1202, E-MEXP-121 and data in [45]). Each dataset was based on Affymetrix micro-arrays. Details can be found at S1–S3 Tables (Sheet 2). Raw probe set signal intensities were normalized independently, using the frozen RMA method (fRMA Bioconductor R package [98]. Individual expression matrices were merged and the residual technical batch effects were corrected using the ComBat method implemented in the SVA R package [99]. Samples corresponding to tumour biopsies with validated presence/absence of *PAX3-FOXO1* or *PAX7-FOXO1* fusion genes were subset from the original data using a custom made R script. Differential analysis of fusion positive versus negative samples was conducted using the samr package and the following parameters: resp.type = "Two class unpaired", nperms = 100, random.seed = 37, testStatistic = "standard"; [100]. Genes with a delta score lower than 2.3 (FDR 0) where selected for subsequent analysis.

Hierarchical clustering of the normalized transcriptomes was implemented using the heatmap.2 function from the gplot package [101]). PAX3-FOXO1 chIPseq data (GSE19063, Cao 2010) were mapped to human genome (hg19) using Bowtie2 [102] and peaks were called using MACS2 [103] implemented on Galaxy server [104]. Peaks common to the 2 replicates of Rh4 cell line and not present in the RD cell line samples were selected using BEDtools [105] and annotated to the two nearest genes using GREAT [106]. Functional annotation of the differentially expressed genes and the PAX3-FOXO1 putative target genes was made using the analysis tool of the PANTHER Classification System [107] or GSEA [108].

### *Chick in ovo* electroporation

Electroporation constructs based on *pCIG* (*pCAGGS-IRES-NLS-GFP*) expression vector [109] have been described previously; *Pax3*, *Pax7*, *PAX3-FOXO1*, *PAX7-FOXO1* [80]; *MYCN* [68]; *CCND1* [110]. Reporters for the human or the mouse versions of PAX3-FOXO1 bound enhancers were cloned upstream of the *thymidine kinase* (tk) promoter and *nuclear LacZ* [111] or *adenovirus major late promoter* (mlp) and *H2B-Turquoise*. For detailed cloning strategies see supporting methods. Reporter plasmids (0.5μg/μl) and *pCIG* based constructs (1.5–2 μg/μl) were electroporated in Hamburger and Hamilton (HH) stage 10–11 chick embryos according to described protocols [112]. Embryos were dissected at the indicated stage in cold PBS 1X.

### Immunohistochemistry and *in situ* hybridisation on cryosections

Embryos were fixed with 4% paraformaldehyde (PFA) for 45 min to 2 hr at 4˚C, cryoprotected by equilibration in 15% sucrose, embedded in gelatin, cryosectioned (14 μm), and processed for immunostaining [112] or *in situ* hybridisation (ISH) [113]. Details of the reagents are provided in the S1 Methods. Immunofluorescence microscopy was carried out using a Leica TCS SP5 confocal microscope. Pictures of *in situ* hybridisation experiments were then taken with an Axio Observer Z1 microscope (Zeiss). All the images were processed with Image J v.1.43g image analysis software (NIH) and Photoshop 7.0 software (Adobe Systems, San Jose, CA, USA). All quantifications were performed using ImageJ v.1.43 g on usually more than 3

embryos and on 2 to 4 transverse sections per embryo. The number of cells positive for a marker per section was established using the cell counter plugin on between 2 to 6 transverse sections per embryo. The number of sections taken into account per embryo varies as a function to the extent of electroporated cells found along the anterior-posterior axis of the embryo. The mean value per embryo has been calculated and is represented with a dot on graphs. Fluorescence intensities in GFP$^+$ cells were determined using ellipsoid region of interests whose size was adapted to that of cell nuclei and multi-measurement plugin. These values were measured at least in 3 embryos and often in more than 5 embryos, the number of embryos analysed is always given in the figure legend. For all intensities, the greatest variations in the data was set between cells and not between embryos. This is likely due to the developmental stage of each cells, their localisation within the neural tube and the levels of expression of the PAX variants. All probabilities of similarity between two populations of values (i.e. between two types of chick samples) were evaluated using a Mann-Whitney U test in GraphPad Prism and all the p-values are given in figures legend. All quantified data can be found in S5 Table.

### EdU pulse labelling and staining

A solution of EdU 500uM was injected within the neural tube lumen 20h before harvest. Immunofluorescence and EdU staining was performed as described previously [114] and with the Click-it EdU system (Thermo fisher).

### Cell dissociation from chick embryos

GFP positive neural tube regions were dissected after a DispaseI-DMEM/F-12 treatment (Stem cell technologies 1U/ml #07923; 37˚C, 30min). Single cell suspensions were obtained by 3 minutes incubation in Trypsin-EDTA 0.05% (Life technologies) and mechanical pressure. Inhibition of Trypsin was ensured using with cold foetal bovine serum (FBS).

### RT-quantitative real-time-PCR on FAC sorted cells

GFP$^+$ cells were sorted using BD Influx Sorter (BD Biosciences). Total RNA was extracted from 60 000 to 80 000 cells following RNAqueous-Micro kit with DNAseI (Life technologies) instructions. RNA quality was assessed by spectrophotometry (DeNovix DS-11 FX spectrometer). cDNA was synthesized by SuperScript VILO (Life Technologies) according to manufacturer's instructions. RT–PCR was performed using the Veriti ™ 96- Well Fast Thermal Cycler (Applied Biosystems) and real-time qPCR was performed with the StepOnePlus™ real-time PCR system (Applied Biosystems) using SYBR Green detection tools (Applied Biosystems). Primers can be found in S1 Methods. The expression of each gene was normalised to that of *TBP*. *ALK*, *CDC42*, *CDH3*, *CDK2*, *FOXF1*, *MYOD1*, *PITX2*, *RB1*, *TFAP2α*, *TFAP2β*, *TBP* expressions were assessed in n = 6 (*pCIG*); n = 4 (*Pax3*); n = 6 (*PAX3-FOXO1*) independent experiments. Other genes were tested in 3 independent experiments per condition. Data representation and statistical analyses using Mann-Whitney U-test or two-way ANOVA test were performed in GraphPad Prism.

### Flow cytometry-based cell cycle analysis

Dissociated cells were stained with 5uM Vybrant DyeCycle violet stain (V35003, Thermo Fisher) and Hoechst 33342 at 37˚C for 30 minutes in the dark. Light scattering parameters were quantified using a Cyan ADP flow cytometer (Beckman Coulter, USA). Data were processed using Flowjo software v10.7.1 (Becton Dickinson, USA). Representative gating strategy is presented in S6A Fig. Single cell events were gated by forward scatter (FCS) peak vs Area

(S6Ai Fig). Cells were also selected based on their granularity FSC Area vs Side Scatter (S6Aii Fig). FSC arear vs GFP-log properties were used to segregate GFP$^+$ electroporated or transfected cells from GFP$^-$ wild-type cells (S6Aiii Fig). Cell cycle analysis was performed by using the Dean/Jett/Fox model for univariate DNA content data with manual constraining G1 and G2 range for model fit optimisation (S6B Fig). Graphs giving the percentage of cells in each cell cycle phase was generated using Excel or GraphPad (Fig 6C; S6C Fig).

### GFP and DNA labelling and imaging 3D chick embryos

Samples were incubated overnight with Atto488 (1/300, Sigma) at 4°C for GFP staining, washed thoroughly in PBS, incubated 5–10 minutes in DRAQ5 (1/1000, Thermofisher) for DNA staining and finally washed in PBS. Samples were mounted on their ventral side in 1% agarose for 3D imaging. 3D scans of samples were obtained with a 2-photon microscope LaVision equipped with a femtosecond pulsed Insight Spectra Physics laser, a Carl Zeiss 20x, NA 1.0 (water immersion) objective and the InSight (LaVision BioTek) image acquisition software. A single wavelength of 930nm was used for exciting all fluorophores to avoid drift artefacts. Two GaAsp sensitive photomultipliers allowed simultaneous detection of the two emission lights which were segregated thanks to a dichroic mirror 585nm and a bandpass filter 525/50nm.

### 3D images processing and quantitative analyses

Image pre-processing and segmentation were performed using ImageJ and Imaris. Background subtraction was performed on GFP channel to eliminate autofluorescence coming from the tissue. Bleach correction normalizing the brightness of images along tissue thickness was performed on DRAQ5 channel stacks of thick samples, notably 72hpe samples. A quality filter on the Imaris automatic surface segmentation plugin based on intensity and size (<95 voxels) allowed removal of saturated unspecific objects and dead cells x,y,z coordinates of the centre point, the major axis of their ellipsoid fit, the sphericity and prolate ellipticity were retrieved for all segmented nuclei. The surfaces encompassing the neural tube, the neural cavity, pCIG progenitors and neurons were delineated on the DRAQ5 signal, on x-y planes every 3 z-stacks. Distance Transformation plugin outside neural cavity segmentation was used to quantify the distance between the centre of the nuclei and this cavity. *Cell clustering* was studied by running DBSCAN algorithm on Matlab [115]. Clusters contained a minimum of 3 cells, and the minimal distance between two cells that belong to the same cluster was fixed to 10μm. *Cell orientation* was established by converting the cartesian coordinates of the vector representing the major axis of the ellipsoid fit of GFP positive cells into polar coordinates (S2C Fig) using Matlab. Matlab or Graphpad Prism were used for graphic representation and statistical analyses.

### Imaging the apical surface of Par3 and GFP labelled spinal cord

Dissected spinal cords were fixed in PFA4% for 1h and washed in PBS. Immunofluorescences were performed on cryosections. Open-book preparation of the samples flatten between a slide and coverslip was imaged using a spinning disk confocal microscope (Leica DMi8: CSU-W1 Yokogawa spinning disk) and MetaMorph (Molecular Devices) image acquisition software.

## Supporting information

**S1 Fig.  (A) Body locations of RMS biopsies.** Locations of FP-RMS (red) and FN-RMS (blue) biopsies whose transcriptome has been assessed in Fig 2A and coming from previous studies

[42–46]. ND: Non determined. **(B) PITX2 expression distinguishes FP-RMS from FN-RMS cells. (i, iii)** Pictures of western blots using the indicated antibodies on proteins extracted from the indicated FN-RMS (RD, RDAbl, Rh36) and FP-RMS (Rh3, Rh5, SJRH30, Rh4) cell lines and **(ii)** normalized PAX3-FOXO1 levels to that of GAPDH. This shows variable levels of PAX3-FOXO1 (i, ii) between FP-RMS cell lines and of MYOD1 across all RMS cell lines (iii). In addition, specific ectopic expression of several PITX2 isoforms (iii) in FP-RMS versus FN-RMS cell lines is revealed (see also S1 to S4 Raw images).
(TIF)

**S2 Fig. Extended characterization of PAX3-FOXO1's ability to induce FP-RMS signature genes. (A) (i-ii')** Immunodetection of GFP and PAX2 on transverse sections of chick embryos 48hpe with the indicated plasmids. **(iii)** Quantification of PAX2 expression levels in GFP$^+$ cells in the spinal cords of chick embryos 48hpe with the indicated constructs (dots: single cell values; bars: mean ± s.e.m.; n>5 embryos). **(B)** Immunodetection of GFP and MYOG on transverse sections of chick embryos 48hpe with the indicated plasmids. x" panels are views on the myotome (myo) of x panel embryo. **(Ci-iii', D)** Immunostaining for GFP, Turquoise direct fluorescence and DAPI staining on transverse sections of chick embryos 24hpe with the indicated plasmids and a reporter for human *PRDM12$^{CRM}$* and *CDH3$^{CRM}$*. **(Civ)** Quantification of Turquoise levels normalised to that of GFP in cells electroporated with *PRDM12$^{CRM}$* reporter at 24hpe (dots: single cell values; bars: mean ± s.e.m.; n>4 embryos). Mann-Whitney U test p-value: ****: p< 0.0001. Scale bars: 50μm.
(TIF)

**S3 Fig. Cell shape and orientation dynamics induced by PAX3-FOXO1. (A) (i-iii)** Projection of 3D images of embryos 48hpe with the indicated plasmids, stained with DRAQ5 and immunolabelled for GFP. **(i'-iii')** Result of the segmentation performed at the level of the boxes indicated on samples i-iii. Surfaces delimiting the electroporated half of the neural tube are transparent yellow, while cell nuclei are coloured. In *pCIG* sample, the surface segregating progenitor nuclei from neurons is highlighted in transparent red. **(iv)** x (medial-lateral), y (antero-posterior) and z (dorsal-ventral) axes giving the orientation of i-iii samples. **(B) (i-iii)** Representative 3D shape of GFP$^+$ nuclei segmented from scanned whole embryos 48hpe with the indicated plasmids. **(iv)** Temporal dynamics of the ellipticity of nuclei measured from the segmentation of GFP$^+$ nuclei (as shown in i-iii) in whole-mount embryos (mean ± s.d., n>6 embryos). **(C) (i)** Representation in the 3 dimensions of the chick embryos of θ and φ polar angles of the vector (blue arrow) defining the major axis of a cell ellipsoid fit (blue circle). **(ii-iii)** φ (ii) and θ (iii) possible values and major axes of chick embryos (black circles) and measured values in embryos electroporated with the indicated plasmids at 48hpe.
(TIF)

**S4 Fig. Extended characterization of the epithelial-mesenchymal transition triggered by PAX3-FOXO1. (A)** Immunodetection of GFP, PARD3, activated βCATENIN (βCAT.) and β1-INTEGRIN on transverse sections of chick embryos 48hpe with the indicated plasmids. In i and ii, x' and x" panels are blown up on a subset of x panel GFP$^+$ cells; in iii and iv the x and x" are blown up on a subset of x panel GFP$^+$ and GFP$^-$ cells. Arrowheads in x' panels point are the accumulation of β1-INTEGRIN on the basal side of cells. Arrows in iii' indicate increased levels of β1-INTEGRIN at the membrane of GFP$^+$ cells. bl: basal lamina. Scale bars: 50μm. **(B)** Normalized levels of *SNAI1* and *ADAM10* mRNA assayed by DNA microarrays in FP-RMS and FN-RMS biopsies (dots: RNA sample values; bars: mean ± s.e.m.; Mann-Whitney U test p-value: ****: p<0.0001).
(TIF)

**S5 Fig. Cell cycle state of PAX3 and PAX3-FOXO1 overexpressing embryonic spinal cells. (A) (i-iii")** GFP and activated CASPASE3 immunodetection and DAPI staining on transverse section of chick embryos 48hpe with the indicated plasmids. **(iv)** Quantification of the number of activated CASPASE3$^+$ cells in the GFP$^+$ population in embryos 48hpe with the indicated plasmids (dots: embryo values; bar plots: mean ± s.e.m.). **(B)** GFP and MCM2 immunodetection and DAPI staining on transverse sections of chick embryos 48hpe with the indicated plasmids. **(iv)** Quantification of MCM2 levels in the GFP$^+$ cells 48hpe with the indicated plasmids (dots: single cell values; bar plots: mean ± s.e.m.; n>5 embryos). **(C)** Heatmaps indicated fold changes in the expression of the indicated genes relative to their mean expression in *pCIG* samples assayed in FAC sorted GFP$^+$ from chick embryos 48hpe with the indicated constructs. **(D)** Quantification of the number of PH3$^+$ cells in the GFP$^+$ population in embryos 24hpe with the indicated plasmids (dots: embryo values; bar plots: mean ± s.e.m.). x' and x" panels are blown up on a subset of x panel GFP$^+$ cells. Mann-Whitney U test p-value: *: p< 0.05, **: p<0.01, ***: p<0.001, ****: p<0.0001, ns: p>0.05. Scale bars: 50μm.
(TIF)

**S6 Fig. Cell cycle phases of PAX3 and PAX3-FOXO1 overexpressing cells. (A)** FACS gating strategy in 3 steps using Flowjo: (i) isolation of singlets (FS: forward scatter/approximation of cell size; lin: linear); (ii) selection of cells based on their size (FS: forward scatter) and granularity (SS: side scatter); (iii) segregation of GFP$^+$ from the GFP$^-$ pools. **(B)** FACS plots showing DNA content distribution of GFP$^-$ (i,ii,iii) and GFP$^+$ (i',ii',iii') chick neural cells stained with Vybrant DyeCycle Violet stain (black line) and the Dean/Jett/Fox model based cell cycle phases gating (pink line: extrapolation of the distribution with the model; pink area: G0/G1 phase, blue area: S phase and green area: G2/M phase) in chick embryos 48hpe with the indicated plasmids. **(C)** Proportion of cells in the indicated cell cycle phase assayed by DNA content distribution of FAC sorted GFP$^-$ and GFP$^+$ chick neural (i-iii) and Human Forskin Fibroblats (HFF; iv-vi) stained with Vybrant DyeCycle Violet stain (dots: mean value on cells analysed on independent FAC sorted samples; bar plots: mean ± s.e.m., unpaired test p-value: *: p< 0.05, **: p<0.01, ***: p<0.001, ns: p>0.05).
(TIF)

**S7 Fig. Extended characterization of PAX7-FOXO1 transformation properties. (A) (i-ii")** Immunodetection of GFP and SOX2 on transverse sections of chick embryos 48hpe with the indicated plasmids. MZ: Mantle Zone; SVZ: Sub-Ventricular Zone. **(iii)** Percentage of SOX2$^+$ cells in the GFP$^+$ population 48hpe with the indicated plasmids (dots: embryo values; bar plots: mean ± s.e.m. **(B)** *In situ* hybridization for *LMO4* (i-ii"), *PITX2a* (iii-iv") and immunodetection of GFP and PAX2 (v-vi") on transverse sections of chick embryos 24hpe (i-iv") or 48hpe (v-vi') with the indicated plasmids. x" panels in i-iv in display region of the DRG, somite or endoderm regions of x sample. DRG: dorsal root ganglia; endo: endoderm; myo: myotome. **(C)** Quantification of the number of PH3$^+$ cells in the GFP$^+$ cells in embryos 48hpe with the indicated plasmids. **(D)** Immunodetection of GFP and CDKN1c on transverse sections of chick embryos 48hpe with the indicated plasmids. P: progenitors; N: neurons. x' and x" panels are blown up on a subset of x panel GFP$^+$ cells. Mann-Whitney U test p-values: *: p<0.05, **: p< 0.01, ****: p<0.0001, ns: p>0.05. Scale bars: 50μm.
(TIF)

**S1 Methods. Enhancer reporter cloning steps, Primers and antibodies lists.**
(DOCX)

**S1 Table. Gene expression levels in FP-RMS biopsies.** Normalised expression levels of genes assayed using DNA-microarrays in FP-RMS biopsies (See Methods)
(XLSX)

**S2 Table. Gene expression levels in FN-RMS biopsies.** Normalised expression levels of genes assayed using DNA-microarrays in FN-RMS biopsies
(XLSX)

**S3 Table. Origin of transcriptomes presented in S1 and S2 Tables and location of PAX3-- FOXO1 bound regions nearly the genes assayed.** Sheet 1: Origin of the samples and presence or not of PAX3-FOXO1 or PAX7-FOXO1. Sheet 2: Identity of the PAX3-FOXO1 bound CRM (peaks) nearby the genes assayed in Sheet1. Sheet 2: Position of PAX3-FOXO1 bound CRM (peaks) on Hg19 genome.
(XLSX)

**S4 Table. Results of the Gene Ontology Biological Process term enrichment analysis.** Sheet 1: Statistics for GO terms related to cell identity, migration and cell cycle regulation enriched in FP-RMS compared to FN-RMS. Sheet 2: FP-RMS upregulated genes assigned to cell identity. Sheet 3: FP-RMS upregulated genes assigned to cell migration and adhesion. Sheet 4: FP-RMS upregulated genes assigned to cell cycle regulation
(XLSX)

**S5 Table. Matrices of the data graphed in the manuscript Figures.**
(XLSX)

**S6 Table. Cell parameters quantified from 3D scans of whole embryos at 48hpe.** Data obtained on a given scan on a given embryo are presented in one independent sheet. Data obtained for a given cell is presented on a single line.
(XLSX)

**S1 Raw image. Full western blot membrane presented in S1B Fig_anti-FOXO1.**
(TIF)

**S2 Raw image. Full western blot membrane presented in S1B Fig_anti-GAPDH.**
(TIF)

**S3 Raw image. Full western blot membrane presented in S1B Fig_anti-PITX2.**
(TIF)

**S4 Raw image. Full western blot membrane presented in S1B Fig_anti-MYOD1.**
(TIF)

## Acknowledgments

We deeply thank the ImagoSeine core facility of Institut Jacques Monod, a member of France-BioImaging (ANR-10-INBS-04) and certified IBiSA. We thank Griselda Wentzinger and Magali Fradet for performing cell sorting at ImagoSeine Institut Jacques Monod platform. We are grateful to the people who have provided us with useful tools. We are grateful to Roger Karess for his English wording tips. We received plasmids from Sophie Bel Vialar, Marie Henriksson, Elisa Marti, Gwen Le Dréau and YiPing Chen and FP-RMS and ERMS cell lines from Cécile Gauthier-Rouvière.

## Author Contributions

**Conceptualization:** Gloria Gonzalez Curto, Audrey Der Vartanian, James Briscoe, Pascale Gilardi-Hebenstreit, Vanessa Ribes.

**Formal analysis:** Gloria Gonzalez Curto, Audrey Der Vartanian, Youcef El-Mokhtar Frarma, Line Manceau, Lorenzo Baldi, Selene Prisco, Vanessa Ribes.

**Funding acquisition:** Frédéric Relaix, James Briscoe, Vanessa Ribes.

**Investigation:** Gloria Gonzalez Curto, Audrey Der Vartanian, Youcef El-Mokhtar Frarma, Line Manceau, Lorenzo Baldi, Selene Prisco, Frédéric Causeret, Muriel Rigolet, Vanessa Ribes.

**Methodology:** Gloria Gonzalez Curto, Audrey Der Vartanian, Lorenzo Baldi, Frédéric Causeret, Daniil Korenkov, Vincent Contremoulins, Orestis Faklaris, Vanessa Ribes.

**Project administration:** Vanessa Ribes.

**Resources:** Nabila Elarouci, Frédéric Aurade, Aurélien De Reynies.

**Software:** Gloria Gonzalez Curto, Line Manceau, Lorenzo Baldi, Nabila Elarouci, Vincent Contremoulins.

**Supervision:** Frédéric Aurade, Aurélien De Reynies, Orestis Faklaris, Vanessa Ribes.

**Validation:** Audrey Der Vartanian, Vanessa Ribes.

**Visualization:** Vanessa Ribes.

**Writing – original draft:** Vanessa Ribes.

**Writing – review & editing:** Gloria Gonzalez Curto, Audrey Der Vartanian, Line Manceau, Frédéric Causeret, Frédéric Relaix, James Briscoe, Pascale Gilardi-Hebenstreit.

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
