## [Decision Letter · Decision Letter 0]

17 Jun 2020

Dear Dr Ribes,

Thank you very much for submitting your Research Article entitled 'The PAXFOXO1s trigger fast trans-differentiation of chick embryonic neural cells into alveolar rhabdomyosarcoma with tissue invasive properties limited by S phase entry inhibition' to PLOS Genetics. Your manuscript was fully evaluated at the editorial level and by independent peer reviewers. The reviewers appreciated the attention to an important problem, but raised some substantial concerns about the current manuscript. Based on the reviews, we will not be able to accept this version of the manuscript, but we would be willing to review again a much-revised version. We cannot, of course, promise publication at that time.

If you decide to revise the manuscript for further consideration at PLOS Genetics, please aim to resubmit within the next 60 days, unless it will take extra time to address the concerns of the reviewers, in which case we would appreciate an expected resubmission date by email to plosgenetics@plos.org.

[LINK]

We are sorry that we cannot be more positive about your manuscript at this stage. Please do not hesitate to contact us if you have any concerns or questions.

Yours sincerely,

Gerard Cornelis Grosveld, Ph.D.

Guest Editor

PLOS Genetics

David Kwiatkowski

Section Editor: Cancer Genetics

PLOS Genetics

Reviewer's Responses to Questions

**Comments to the Authors:**

Reviewer #1: In this report, the authors study the effect of the PAX3-FOXO1 and PAX7-FOXO1 chimeric proteins in embryonic chick neural tube. This model allows the authors to study the primary events that are driven by these oncogenic proteins, prior to the development of tumors. The authors found that the expression of PAX3-FOXO1 and PAX7-FOXO1 is sufficient to alter the transcriptional program of neural cells, which adopt a molecular signature associated with ARMS cells. In addition, they found that the fusion proteins promote cell migration and tissue invasion in their model. Finally, they also observe that PAX3/7-FOXO1 blocks cell cycle progression, providing one molecular explanation as to why these oncogenic fusions are inefficient in mediating cell transformation. Overall, the study is interesting and adds to our understanding of the early steps of cellular transformation by PAX3/7-FOXO1. Although it is well established that different oncogenes induce cell cycle arrest, this has not previously been shown for ARMS. Unfortunately, the manuscript itself is very hard to read, sometimes unclear, and the relevance of all the different figure panels is not always justified. The figures could be re-arranged to improve the clarity of the manuscript and to follow the text better. There is also some inconsistency between the different figure panels. Overall, the authors findings are interesting but the paper itself needs clarifications, re-writing and more cohesion.

Major points:

1. Throughout the paper, all the figure panels should display a) examples of immunostaining; and b) quantification; for all the different conditions tested (empty vector, PAX3, PAX7, P3-FOXO1 and P7-FOXO1). Here are some specific examples:

- Figure 1: show immunostaining for the empty vector

- Figure 1: quantification (v, vi, vii) needs to be done for PAX7 and PAX7-FOXO1

- Figure 1C: No panel for PAX6 staining in the PAX7-FOXO1 cells

- Figure 2F: please show the staining for PAX3 alone (and possibly add PAX7 and PAX7-FOXO1)

- Figure S1C: top panel shows TFAP2 quantification for PAX7 and PAX7-FOXO1, but in the bottom panel PAX7 and PAX7-FOXO1 are absent from the PAX2 quantification

- Figure 3: the authors tested only one enhancer for the PAX7-FOXO1 fusion

- etc…

2. The authors elegantly combined and re-analyzed microarrays from different ARMS studies in Table S1 and S2, which provide a rich resource for anyone aiming at identifying key molecular events taking place in these tumors. However, in Figure 2, the authors opted to focus on a set of transcription factors which expression is elevated in ARMS. Are these TFs the best and more relevant markers for ARMS? What about the set of genes that are normally exclusively expressed in the myogenic lineage (in addition to MYOD)? Or what about the genes that display the highest up-regulation in ARMS, irrespective of their function?

Minor points:

1. Does the expression of PAX3-FOXO1 or PAX7-FOXO1 drive oncogene-induced senescence? It may be out of the scope of the study, but that would strengthen the paper.

2. In Figure 2, the authors should disclose which proportion of the ARMS samples display PAX3-FOXO1 fusion, PAX7-FOXO1 fusion or no fusion. This information can be found in Supplemental Table S3, but it should appear in the manuscript.

3. In Figure 2, the authors analyze the expression of EYA2, but its relevance is not specified in the text. Please edit and add the appropriate references.

4. In Figures 2Dv and 2Ev, the authors should indicate which genes are statistically and significantly different in ARMS (2Dv) and in PAX3-FOXO1 (2Ev), for example by using an asterisk.

5. In Figure 2E, it would be interesting and relevant to determine whether these sets of genes are also elevated in PAX7-FOXO1 fusion.

6. In Figure S1B, the authors should perform additional immunoblottings for the different genes they highlight in Figure 2, in addition to PITX2. These additional Western blots should include MYOD.

7. It looks like throughout the paper, the statistics are calculated using a n number that corresponds to the number of cells analyzed. The n number employed for statistical tests should correspond to the number of biological replicates (i.e. number of embryos).

8. In the text, the authors state that “Conversely, forced expression of PAX3 or PAX7 tented to reduce the number of TFAP2+ and PAX2+ neurons generated ». This is not true for TFAP2. As for PAX2, the data is not shown for the PAX7 condition. For these reasons, the authors statement is not justified.

9. In Figure 2F, the authors should add a staining for MYOD, a gene that is normally expressed exclusively in myogenic cells, unlike LMO4 and PITX2.

10. It looks like the same dataset is presented in Figure 5Aiii and S3Av. Please make the appropriate corrections in the revised manuscript.

11. In Figure 5B, please specify which plasmid was electroporated in the embryo from which the representative immunostaining picture was selected (5Bi and 5Bii).

12. In Figure 5C, the authors should include a representative flow cytometry plot (cell cycle) for all the different conditions. In addition, it is not clear in the Methods section how the proportions of cells in the different cell cycle phases was calculated. Separating the flow cytometry plot into 3 bins is not appropriate to assess the distribution into cell cycle phases. Many softwares are available to analyze cell cycle profiles, for example ModFit LT. The authors should perform cell cycle analyses with an appropriate software and clearly indicate it in the Methods section.

Reviewer #2: In this manuscript the authors extensively utilize an in ovo model to follow the response of spinal cord progenitors to PAX3/7-FOXO1. As expected, they show that PAX3/7-FOXO1 but not wild-type PAX3 and PAX7, transform these cells down a myogenic lineage resembling ARMS. The cells show mesenchymal and tissue invasion properties. They show that PAX3/7-FOXO1 and wild-type PAX3/7 reduces the levels of CDK-CYCLIN activity and arrests cells in G1. They further show, as predicted, that CYCLIN D1 or MYCN overcomes this cell cycle inhibition and promotes tumor growth.

The manuscript is well written, and the body of work represents an additional and novel model using chick embryos for investigating the oncogenic function of the PAX3/7-FOXO1 fusion oncogene. The data is extensive and robust.

There are a few typos in the manuscript that the authors should correct. For example page 9 “Conversely, forced expression of PAX3 or PAX7 tented to reduce …” should be “tended to reduce..”.

Reviewer #3: Manuscript Review: PLOS GENETICS ID# - PGENETICS-D-20-00825

Title: The PAXFOXO1s trigger fast trans-differentiation of chick embryonic neural cells into alveolar rhabdomyosarcoma with tissue invasive properties limited by S phase entry inhibition.

The conventional forms of treatment available to rhabdomyosarcoma (RMS) patients are limiting, and too often not effective for patients with advanced or relapsed cases. Thus, emphasizing the need to identify a novel targeted therapeutic approach by which to treat RMS patients, the authors in this interesting manuscript are reporting an in ovo chick model for PAX-FOXO1-driven pathobiology. The authors demonstrate that PAX-FOXO1 (abbreviation for both PAX3- and PAX7-FOXO1) drives ectopic myogenesis in avian neural tube, while wild-type PAX3 is minimally-to-completely ineffective. Further interrogation shows gene signatures consistent with PAX-FOXO1-positive RMS, but not fusion-negative (embryonal) RMS. The authors also identify interesting targets of the PAX-FOXO1 transcription factor, and verify that the targets identified in their avian model correlate with human RMS. The data provided are of high quality and convincing.

This reviewer is torn, however, as the studies presented in this manuscript, regretfully, cannot be described as completely novel. The reason for this assessment is described as follows.

Major Considerations:

1) My only “Major Consideration” for this manuscript is that the ability of PAX3 and/or PAX3-FOXO1 to transform chick neural progenitor tissue into muscle is not unique, as at least two such published manuscripts dated ~15-25 years ago report similar findings.

In Maroto et al, Cell (1997), the authors elegantly used avian neural tube explant primordia to show that ectopic wild-type PAX3 transforms these non-muscle precursors into muscle-lineage cells. Thus, the notion that the PAX3 DNA-binding domains can drive ectopic myogenesis in avian neural precursors is known.

In Finkenstein et al, Transgenic Res (2006), the authors report that transgenic mice expressing PAX3-FOXOI show multiple defects in muscle development, including ectopic myogenesis in the embryonic neural tube.

In tandem, these two studies have already reported the main selling-points to this manuscript’s primary potential impact. Thus, even though the authors build nicely on these previous models given modern-day technology, the manuscript reads more as a modern-day extension rather than a novel study in the RMS field.

Of note, both of these studies are not referenced in the manuscript, which is a very important missing component. This reviewer believes that the authors need to explain in the manuscript how their studies are significantly novel in light of these prior studies. A convincing discussion on the issue could alter my perception.

I do want to restate, however, that I really enjoyed reading this manuscript and note the excellent quality of the data, especially the tissue fluorescence studies

Minor Considerations:

1) Of note, the RMS field has essentially ceased using the term Alveolar RMS or Embryonal RMS. Instead, the WHO classifications are now Fusion-Positive (PAX3- or PAX7-FOXO1-positive) RMS and Fusion-Negative RMS.

2) Though the data and figures are of high quality, the figures are very, very dense to examine and process. The authors are advised to contemplate more reader-friendly configurations for their copious amount of data.

**Have all data underlying the figures and results presented in the manuscript been provided?**

Reviewer #1: Yes

Reviewer #2: Yes

Reviewer #3: Yes

PLOS authors have the option to publish the peer review history of their article (what does this mean?). If published, this will include your full peer review and any attached files.

Reviewer #1: No

Reviewer #2: Yes: Javed Khan

Reviewer #3: No

---

## [Decision Letter · Decision Letter 1]

9 Sep 2020

Dear Dr Ribes,

Thank you very much for submitting your Research Article entitled 'The PAXFOXO1s trigger fast trans-differentiation of chick embryonic neural cells into alveolar rhabdomyosarcoma with tissue invasive properties limited by S phase entry inhibition' to PLOS Genetics. Your manuscript was fully evaluated at the editorial level and by independent peer reviewers. The reviewers appreciated the attention to an important topic but identified some aspects of the manuscript that should be improved.

We therefore ask you to modify the manuscript according to the review recommendations before we can consider your manuscript for acceptance. Your revisions should address the specific points made by each reviewer.

[LINK]

Yours sincerely,

Gerard Cornelis Grosveld, Ph.D.

Guest Editor

PLOS Genetics

David Kwiatkowski

Section Editor: Cancer Genetics

PLOS Genetics

The reviewers state that the revised manuscript is a substantial improvement over the original submission. However, reviewers 1 and 2 bring up minor issues that need to be addressed in the second revision of the manuscript.

Reviewer's Responses to Questions

**Comments to the Authors:**

Reviewer #1: In this revised version, Gonzalez Curto et al addressed all the concerns raised previously. Their rebuttal letter is very thorough, they took the time to provide a substantial point-by-point response addressing all the issues raised by the different reviewers.

They rearranged their figure panels and simplified the text. They opted to focus on PAX3-FOXO1, and they provided the different controls that were missing in the previous submission.

My only comment for the authors is that the Summit software is not designed and not appropriate to adequately measure cell cycle profiles. The use of a flow cytometry DNA analysis software like ModFit LT (or other) really makes a difference when it comes to cell cycle profile analyses. It is more accurate than simply separating your cells into three bins, as the software models the cells distribution into the different cell cycle phases. I recommend that the authors re-analyze their cell cycle data using such a software.

Reviewer #2: The revised manuscript has addressed a lot of the concerns bought up by the original review.

Several concerns remain.

The conclusion in the abstract “Together, our findings reveal a mechanism underpinning the apparent limited oncogenicity of PAXFOXO1 fusion transcription factors and support a neural origin for FP-RMS.” is too strong given the lack of evidence provided by the authors of the neural origin, and the overwhelming evidence that the cell of origin is likely to be muscle progenitor cells for the majority of cases. The authors can say that their results show that “PAX3FOXO1 is sufficient to divert cells from a generic neurogenic program” not that it definitively proves a neural origin, given that the PAX3-FOXO1 fusion gene is a pioneering transcription factor able to divert almost any cell into a myogenic lineage, and also able to induce some neural genes given PAX3 is expressed in neural crest that the FOXO1 is a strong transactivator of all targets of PAX3.

Other minor concerns:

1.Throughout the manuscript the authors should change PAX3FOXO1 to PAX3-FOXO1 etc., which is the more standard nomenclature.

2.The sentence “Genetically re-activating core cell cycle regulators lift up PAXFOXO1 mediated cell cycle inhibition” is awkward does the author mean rescue PAX-FOXO1 mediated cell cycle inhibition?

3.The authors should cite PMID: 10557309 which was one of the first manuscripts to show the activation of a myogenic transcription program by PAX3-FOXO1.

4.The authors state “ several of the target genes encode cell surface proteins required for FP-RMS cells migration”. Change “required” to “associated with” because it is not clear that these genes are required for migration.

5.The authors state “ FP-RMS cells harbour undifferentiated muscle cell determinants” not sure what this means, please expand.

6.The authors state “ PAXFOXO1s perturb the core cell cycle machinery while PAX3FOXO1 activity during G2/M and G1/S checkpoints facilitates the amplification of transformed cells” I am unsure of the meaning of “facilitates the amplification of transformed cells”.

7.Figure 1 Legend The cervical to thoracic region (blue square) blue looks black not blue.

8.In l Figure 1 Legend specify “hpe” means hours post electroporation.

9.Several cell lines are mentioned which appear to be identical in Figure 2D cell e.g. lines RD and RDabl and RH30 and SJRh30. Please clarify why identical lines are used and add to manuscript.

10.Generally, although the references to figures are better than the original manuscript there still is a poor referencing in the manuscript.

11.The sentence “the histology of the transformed cells strongly support a routing of cells not towards a myoblast like state but a FP-RMS like state” on Page 38 Not sure what the authors mean as there is much evidence that PAX3-FOXO1 does route cells to a myoblastic like state.

12.The statement “This idea is further supported by the observation that the regulatory regions interacting with PAX3 promoter upon t(2;13)(q35;q14) translocations can remain active in the neural tube after the translocation” is ambiguous. What regions is the author is referring to? There is evidence that the FOXO1 region involved in the translocation is also active in myogenesis.

13.For the comment “These results provide insight for why complementation of PAXFOXO1s with genetic aberrations promoting cell cycle progression, such as the gain of MYCN or CCND1, or loss of p53 or RB1, can enhance their tumorigenic potential”, please add CDKN2A loss.

Reviewer #3: I am pleased to note that the authors have satisfactorily addressed my comments, as well as most from the other reviewer. I am happy to now recommend this manuscript for publication in PLOS Genetics.

**Have all data underlying the figures and results presented in the manuscript been provided?**

Reviewer #1: Yes

Reviewer #2: Yes

Reviewer #3: Yes

PLOS authors have the option to publish the peer review history of their article (what does this mean?). If published, this will include your full peer review and any attached files.

Reviewer #1: No

Reviewer #2: No

Reviewer #3: **Yes: **Rene Galindo

---

## [Editor Report · Decision Letter 2]

2 Oct 2020

Dear Dr Ribes,

We are pleased to inform you that your manuscript entitled "The PAX-FOXO1s trigger fast trans-differentiation of chick embryonic neural cells into alveolar rhabdomyosarcoma with tissue invasive properties limited by S phase entry inhibition" has been editorially accepted for publication in PLOS Genetics. Congratulations!

Yours sincerely,

Gerard Cornelis Grosveld, Ph.D.

Guest Editor

PLOS Genetics

David Kwiatkowski

Section Editor: Cancer Genetics

PLOS Genetics

Comments from the reviewers (if applicable):

The authors responded accurately to resolve the remaining points of critique.

**Data Deposition**

http://datadryad.org/submit?journalID=pgenetics&manu=PGENETICS-D-20-00825R2

**Press Queries**

---

## [Editor Report · Acceptance letter]

5 Nov 2020

PGENETICS-D-20-00825R2 

The PAX-FOXO1s trigger fast trans-differentiation of chick embryonic neural cells into alveolar rhabdomyosarcoma with tissue invasive properties limited by S phase entry inhibition 

Dear Dr Ribes, 

We are pleased to inform you that your manuscript entitled "The PAX-FOXO1s trigger fast trans-differentiation of chick embryonic neural cells into alveolar rhabdomyosarcoma with tissue invasive properties limited by S phase entry inhibition" has been formally accepted for publication in PLOS Genetics! Your manuscript is now with our production department and you will be notified of the publication date in due course.

With kind regards,

Kaitlin Butler

PLOS Genetics

On behalf of:
